# Multi-Agent Imitation Learning:
# Value is Easy, Regret is Hard

**Jingwu Tang**
Carnegie Mellon University
jingwutang@cmu.edu

**Gokul Swamy**
Carnegie Mellon University
gswamy@cmu.edu

**Fei Fang**
Carnegie Mellon University
feifang@cmu.edu

**Zhiwei Steven Wu**
Carnegie Mellon University
zstevenwu@cmu.edu

## Abstract

We study a multi-agent imitation learning (MAIL) problem where we take the perspective of a learner attempting to *coordinate* a group of agents based on demonstrations of an expert doing so. Most prior work in MAIL essentially reduces the problem to matching the behavior of the expert *within* the support of the demonstrations. While doing so is sufficient to drive the *value gap* between the learner and the expert to zero under the assumption that agents are non-strategic, it does not guarantee robustness to deviations by strategic agents. Intuitively, this is because strategic deviations can depend on a counterfactual quantity: the coordinator's recommendations outside of the state distribution their recommendations induce. In response, we initiate the study of an alternative objective for MAIL in Markov Games we term the *regret gap* that explicitly accounts for potential deviations by agents in the group. We first perform an in-depth exploration of the relationship between the value and regret gaps. First, we show that while the value gap can be efficiently minimized via a direct extension of single-agent IL algorithms, even *value equivalence* can lead to an arbitrarily large regret gap. This implies that achieving regret equivalence is harder than achieving value equivalence in MAIL. We then provide a pair of efficient reductions to no-regret online convex optimization that are capable of minimizing the regret gap *(a)* under a coverage assumption on the expert (MALICE) or *(b)* with access to a queryable expert (BLADES).

## 1 Introduction

We consider the problem of a *mediator* learning to *coordinate* a group of strategic agents via *recommendations* of actions to take without knowledge of their underlying utility functions (e.g. routing a group of drivers through a road network). Given the difficulty of manually specifying the quality of a recommendation in such situations, it is natural to provide the mediator with data of desired coordination behavior, turning our problem into one of **multi-agent imitation learning** (MAIL, [27, 6, 19, 26, 11]). In our work, we explore the nuances of a fundamental MAIL question:

> *What is the right objective for the learner in a multi-agent imitation learning problem?*

We can begin to answer this question by exploring the following scenario: consider developing a routing application to provide personalized route recommendations ($\sigma$) to a group of users with joint policy $\pi$ (e.g. the routing policy that underlies the recommendations provided in Google Maps [3]). As usual in imitation learning (IL), we assume we are given access to *demonstrations* from an *expert* $\sigma_E$ (e.g. a past iteration of the application). We can imagine two kinds of users of our application (i.e.

38th Conference on Neural Information Processing Systems (NeurIPS 2024).

*agents*): *non-strategic* users who blindly follow the recommendations of our routing application and *strategic* users who will *deviate* from our recommendations if they have the incentive to do so under their (unknown) personal utility function (e.g. we recommend a long route to a busy driver). We use $J_i(\pi_\sigma)$ below to denote the value of the mediator's learned policy $\sigma$ under the $i$th agent's utility.

**Case 1: No Strategic Agents.** In the idealized situation where all agents in the population are perfectly obedient, we can essentially treat a MAIL problem as a *single-agent* IL (SAIL) problem over joint policies. It is therefore natural to use a direct extension of the well-studied **value gap** criterion from the SAIL literature [1, 28, 21, 24, 22, 23, 25, 16] to the multi-agent setting:

$$\max_{i \in [m]} J_i(\pi_{\sigma_E}) - J_i(\pi_\sigma).$$

Intuitively, driving the value gap to 0 (i.e. achieving *value equivalence* in the terminology of [10]) implies that along as long as all agents blindly follow our recommendations, we have learned a policy that performs at least as well as that of the expert from the perspective of *any* agent in the population. In our running routing application example, this means that if no driver deviates from the previous behavior, all drivers will be at least as happy as they were with the prior iteration of the application.

**Case 2: Strategic Agents.** Of course for any MAIL problem where agents actually have *agency*, we need to account for the fact that agents may deviate from our recommendations if it appears beneficial to do so from their subjective perspective. Let us denote the class of deviations (i.e. policy modifications) for agent $i$ as $\Phi_i$. Then, we can define the **regret** induced by the mediator's policy as

$$\mathcal{R}_\Phi(\sigma) := \max_{i \in [m]} \max_{\phi_i \in \Phi_i} (J_i(\pi_{\sigma,\phi_i}) - J_i(\pi_\sigma)),$$

where $\phi_i$ is a strategic deviation of agent $i$ and $\pi_{\sigma,\phi_i}$ is the *joint agent policy induced by all agents other than $i$ following $\sigma$'s recommendations*. Intuitively, regret captures the maximum incentive any agent in the population has to deviate from the mediator's recommendations. We can then compare this metric between the expert and learner policies to arrive at the notion of a **regret gap** [27]:

$$\mathcal{R}_\Phi(\sigma) - \mathcal{R}_\Phi(\sigma_E).$$

Driving the regret gap to zero (i.e. achieving *regret equivalence*) implies that *even if agents are free to deviate, our learned policy is at least as good as the expert's from the perspective of an arbitrary agent in the population*. In our preceding example, this means that despite the fact that they are not forced to follow our application's recommendations, all agents would have no more incentive to take an alternate route than they did under the previous iteration of the application.

A simple decomposition allows us to show that a small value gap does not in general imply a small regret gap. Consider the performance difference between the learner's policy under all obedient ($J_i(\pi_\sigma)$) and a deviating $i$th agent ($J_i(\pi_{\sigma,\phi_i})$). We can decompose this quantity into the following:

$$J_i(\pi_{\sigma,\phi_i}) - J_i(\pi_\sigma) = \underbrace{(J_i(\pi_{\sigma,\phi_i}) - J_i(\pi_{\sigma_E,\phi_i}))}_{\text{(I: value gap under } \phi_i)} + \underbrace{(J_i(\pi_{\sigma_E,\phi_i})) - J_i(\pi_{\sigma_E}))}_{\text{(II: expert regret under } \phi_i)} + \underbrace{(J_i(\pi_{\sigma_E}) - J_i(\pi_\sigma))}_{\text{(III: SAIL value gap)}},$$

where we use $\pi_{\sigma_E,\phi_i}$ to denote agent joint behavior under expert recommendations and deviation $\phi_i$. Term III is the standard single-agent value gap (i.e. the performance difference under the assumption that no agents deviate). Term II is the expert's regret under deviation $\phi_i$ (i.e. a quantity we cannot control). Thus, the difference between the regret gap and value gap objectives can be boiled down to Term I: $J_i(\pi_{\sigma,\phi_i}) - J_i(\pi_{\sigma_E,\phi_i})$. Observe that because of the state distribution shift induced by deviation $\phi_i$, minimizing Term III doesn't give us any guarantees with respect to Term 1. This underlies our key insight: ***regret is hard in MAIL as it requires knowing what the expert would have done in response to an arbitrary agent deviation.*** More explicitly, our contributions are three-fold:

**1. We initiate the study of the *regret gap* for MAIL in Markov Games.** Unlike the value gap – the standard objective in single-agent IL – the regret gap captures the fact that agents in the population may choose to deviate from the mediator's recommendations. The shift from value to regret gap captures what is fundamentally different about the SAIL and the MAIL problems.

**2. We investigate the relationship between regret gap and the value gap.** We show that under the assumption of complete reward and deviation function classes, regret equivalence implies value equivalence. However, we also prove that value equivalence provides essentially no guarantees on the regret gap, establishing a fundamental limitation of applying SAIL algorithms to MAIL problems.

**3. We provide a pair of efficient algorithms to minimize the regret gap under certain assumptions.** While regret equivalence is hard to achieve in general as it depends on counter-factual expert recommendations, we derive a pair of efficient reductions for minimizing the regret gap that operate under different assumptions: MALICE (which operates under a coverage assumption) and BLADES (which requires access to a queryable expert). We prove that both algorithms can provide $O(H)$ bounds on the regret gap, where $H$ is the horizon, matching the strongest known results for the value gap in single-agent IL. See Table 1 for a summary of our regret gap bounds.

|  | Assumption | Upper Bound | (Matching) Lower Bound |
|---|---|---|---|
| J-BC | $\beta$-Coverage | $O\left(\frac{1}{\beta}\epsilon u H\right)$, Theorem 5.1 | $\Omega\left(\frac{1}{\beta}\epsilon u H\right)$, Theorem 5.2 |
| J-IRL | $\beta$-Coverage | $O\left(\frac{1}{\beta}\epsilon u H\right)$, Theorem 5.3 | $\Omega\left(\frac{1}{\beta}\epsilon u H\right)$, Corollary 5.4 |
| MALICE (ours) | $\beta$-Coverage | $O\left(\epsilon u H\right)$, Theorem 5.5 | $\Omega\left(\epsilon u H\right)$, Theorem 5.6 |
| BLADES (ours) | Queryable Expert | $O\left(\epsilon u H\right)$, Theorem 5.7 | $\Omega\left(\epsilon u H\right)$, Theorem 5.8 |

Table 1: A summary of our results: upper and lower bounds on the regret gap (i.e. $\mathcal{R}_\Phi(\sigma) - \mathcal{R}_\Phi(\sigma_E)$) of various approaches to multi-agent IL. Here, $\beta$ is the coverage constant in Assumption 5.2, $u$ is the recoverability constant in Assumption 5.1, $H$ is the horizon.

## 2 Related Work

**Single-Agent Imitation Learning.** Much of the theory of imitation learning focuses on the single-agent setting [14]. Offline approaches like behavioral cloning (BC, [15]) reduce the problem of imitation to mere supervised learning. Ignoring the covariate shift in state distributions between the expert and learner policies can cause *compounding errors* [17, 21] and associated poor performance. In response, interactive IL approaches like inverse reinforcement learning (IRL, [1, 28]) allow the learner to observe the consequences of their actions during the training procedure, preventing compounding errors [21]. However, such approaches can be rather sample-inefficient due to the need to repeatedly solve a hard RL problem [25, 16]. Alternative approaches include interactively querying the expert to get action labels on the learner's induced state distribution (DAgger, [17]) or, assuming full coverage of the demonstrations, using importance weighting to correct for the covariate shift (ALICE, [20]). Our BLADES and MALICE algorithms can be seen as the regret gap analog of the value gap-centric DAgger and ALICE algorithms, operating under the same assumptions.

**Multi-Agent Imitation Learning.** The concept of the regret gap was first introduced in the exceptional work of Waugh et al. [27], though their exploration was limited to Normal Form Games (NFGs), in contrast to the more general Markov Games (MGs) we focus on. Fu et al. [7] briefly consider the regret gap in Markov Games (MGs) but do not explore its properties nor provide algorithms for efficient minimization. Most empirical MAIL work [19, 12, 4, 26, 11] is value gap-based, while we take a step back and ask what the right objective is for MAIL in the first place.

**Inverse Game Theory.** Another line of work focuses on inverse game theory in Markov Games [13, 8], where the goal is to recover a set of utility functions that rationalize the observed agent behavior, rather than learning to coordinate from demonstrations. A detailed comparison between the goals of our work at that of inverse game theory provided in Appendix F.

## 3 Preliminaries

We begin with the notation we will use in our paper. Throughout, we use $\Delta(X)$ denote the space of probability distribution over a set $\mathcal{X}$. We will use $\ell$ to denote the loss function each algorithm optimizes, which should be thought of as a convex upper bound on the total variation distance TV. We use $\ell_{\text{TV}}$ when the loss function is exactly the TV distance.

**Markov Games.** We use $MG(H, \mathcal{S}, \mathcal{A}, \mathcal{T}, \{r_i\}_{i=1}^m, \rho_0)$ to denote a *Markov Game* (MG) between $m$ agents. Here, $H$ is the horizon, $\mathcal{S}$ is the state space, and $\mathcal{A} = \mathcal{A}_1 \times ... \times \mathcal{A}_m$ is the joint action space for all agents. We use $\mathcal{T} : \mathcal{S} \times \mathcal{A} \to \Delta(\mathcal{S})$ to denote the transition function. Furthermore, the reward (utility) function for agent $i \in [m]$ is denoted by $r_i : \mathcal{S} \times \mathcal{A} \to [-1, 1]$. Lastly, we use $\rho_0$ to denote the initial state distribution from which the initial state $s_0 \sim \rho_0$ is sampled.

**Learning to Coordinate.** Rather than considering the problem of learning individual agent policies in the MG, we take the perspective of a *mediator* who is giving recommendations to each agent to help them coordinate their behavior (e.g. a smartphone mapping application providing directions to a set of users). At each time step, the mediator gives each agent $i$ a private *action recommendation* $a_i$ to take at the current state $s$. Critically, no agent observes the recommendations the mediator provides to another agent. We can represent the mediator as a Markovian joint policy $\sigma \in \Sigma$, where $\sigma : \mathcal{S} \to \Delta(\mathcal{A})$. We use $\sigma(\vec{a}|s)$ to denote the probability of recommending joint action $\vec{a}$ in state $s$. We use $\pi : \mathcal{S} \to \Delta(\mathcal{A})$ to denote the joint policy that agents play in response to the mediator's policy. When agents exactly follow the mediator's recommendations, we denote their joint policy as $\pi_\sigma$.

A trajectory $\xi \sim \pi = \{s_h, \vec{a}_h\}_{h=1,\dots,H}$ refers to a sequence of state-action pairs generated by starting from $s_0 \sim \rho_0$ and repeatedly sampling joint action $\vec{a}_h$ and next states $s_{h+1}$ from $\pi$ and $\mathcal{T}$ for $H-1$ time steps. Let $d_h^\pi$ denote the *state visitation distribution* at timestep $h$ following $\pi$ and let $d^\pi = \frac{1}{H} \sum_{h=1}^{H} d_h^\pi$ be the average state distribution. Let $\rho_h^\pi(s_h, \vec{a}_h)$ denote the *occupancy measure* – i.e., probability of reaching state $s$ and then taking action $\vec{a}$ at time step $h$. By definition, we know that $\forall h, \sum_{s,\vec{a}} \rho_h^\pi(s, \vec{a}) = 1$. Let $\rho^\pi(s, \vec{a}) = \frac{1}{H} \sum_{h=1}^{H} \rho_h^\pi(s, \vec{a})$ be the average occupancy measure.

We use $V_{i,h}^\pi$ to denote the expected cumulative reward of agent $i$ under this policy from time step $h$, i.e. $V_{i,h}^\pi(s) = \mathbb{E}_{\xi \sim \pi}[\sum_{t=h}^{H} r_i(s_t, \vec{a}_t)|s_h = s]$. We define Q-value function of agent $i$ as $Q_{i,h}^\pi(s, \vec{a}) = \mathbb{E}_{\xi \sim \pi}[\sum_{t=h}^{H} r_i(s_t, \vec{a}_t)|s_h = s, \vec{a}_h = \vec{a}]$. We define advantage of an agent $i$ to be the difference between its Q-value on a selected action and the V-value on the state, i.e. $A_{i,h}^\pi(s, \vec{a}) = Q_{i,h}^\pi(s, \vec{a}) - V_{i,h}^\pi(s)$. We also define the performance of a policy $\pi$ from the perspective of agent $i$ as $J_i(\pi) = \mathbb{E}_{s_0 \sim \rho_0}[\mathbb{E}_{\xi \sim \pi}[\sum_{t=1}^{H} r_i(s_t, \vec{a}_t)|s = s_0]]$. Observe that performance is the inner product between the occupancy measure and the agent's reward function, i.e. $J_i(\pi) = H \sum_{s,\vec{a}} \rho^\pi(s, \vec{a}) r_i(s, \vec{a})$.

**Correlated Equilibria.** We now introduce the notion of a *correlated equilibrium* (CE, Aumann [2]). First, we define a *strategy deviation* $\phi_i$ for the $i$-th agent as a map $\phi_i : \mathcal{S} \times \mathcal{A}_i \to \mathcal{A}_i$. Intuitively, a strategy deviation captures how the agent responds to the current state of the world and the recommendation of the mediator – they can either obey (in which case $\phi_i(s, a) = a$) or defect (in which case $\phi_i(s, a) \neq a$). Let $\Phi_i$ be the set of deviations for agent $i$, which is a subset of all possible deviations $\phi_i(s, a) \neq a$. We use $\Phi := \{\Phi_i\}_{i=1}^{m}$ to denote deviations for all agents. We assume that for all $i$, the identity mapping $\phi_i(s, a) \equiv a$ is in $\Phi_i$. We use $\pi_{\sigma,\phi_i}$ to denote $(\phi_i \circ \pi_{\sigma,i}) \odot \pi_{\sigma,-i}$: the joint agent policy induced by mediator policy $\sigma$ being over-ridden by deviation $\phi_i$. We can now define a CE.

**Definition 3.1** (Regret and CE in General-Sum MGs). *Let $\sigma \in \Sigma$ be the mediator's policy in a Markov Game, and $\Phi_i$, $i \in [m]$ be the deviation classes for each agent. Then,*

1. *We define the regret of a mediator policy $\sigma$ to be*

$$\mathcal{R}_\Phi(\sigma) := \max_{i \in [m]} \max_{\phi_i \in \Phi_i} (J_i(\pi_{\sigma,\phi_i}) - J_i(\pi_\sigma)), \tag{1}$$

2. *We say a mediator with policy $\sigma$ induces an $\epsilon$-approximate Correlated Equilibrium (CE) if*

$$\mathcal{R}_\Phi(\sigma) \leq \epsilon. \tag{2}$$

Intuitively, regret captures the maximum utility any agent can gain by defecting from the mediator's recommendation. A CE is an induced joint policy where no agent has a large incentive to deviate.

# 4 On the Relationship between the Value Gap and the Regret Gap

As sketched above, we consider two potential objectives for the learner in MAIL:

**Definition 4.1** (Value Gap). *We define the value gap between the expert's policy $\sigma_E$ and the learner's policy $\sigma \in \Sigma$ as*

$$\max_{i \in [m]} (J_i(\pi_{\sigma_E}) - J_i(\pi_\sigma)). \tag{3}$$

**Definition 4.2** (Regret Gap). *We define the regret gap between the expert's policy $\sigma_E$ and the learner's policy $\sigma \in \Sigma$ as*

$$\mathcal{R}_\Phi(\sigma) - \mathcal{R}_\Phi(\sigma_E) = \max_{i \in [m]} \max_{\phi_i \in \Phi_i} (J_i(\pi_{\sigma,\phi_i}) - J_i(\pi_\sigma)) - \max_{k \in [m]} \max_{\phi_k \in \Phi_k} (J_k(\pi_{\sigma_E,\phi_k}) - J_k(\pi_{\sigma_E})). \tag{4}$$

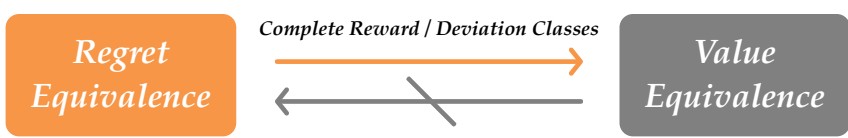

Figure 1: Under expressive enough reward function and deviation classes, regret equivalence implies value equivalence but not vice versa, making the regret gap a "stronger" objective than the value gap.

We say that the learner's policy satisfies *value / regret equivalence* when the value / regret gap is $0$. We now explore the relationship between the value and regret gap in MAIL, [1] summarized in Figure 1. We use $J_i(\pi_\sigma, f)$ and $\mathcal{R}_\Phi(\sigma, f)$ to denote the value/regret of policy $\sigma$ under the reward function $f$.

## 4.1 Regret Equivalence + Complete Reward / Deviation Class $\implies$ Value Equivalence

First, we show that if the reward function class and deviation class are both *complete*, then regret equivalence implies value equivalence. We say that the reward function class is complete when $\mathcal{F} = \{\mathcal{S} \times \mathcal{A} \to [-1, 1]\}$ (i.e. all convex combinations of state-action indicators), and that the deviation class is complete if for every agent $i$, $\Phi_i = \{\mathcal{S} \times \mathcal{A}_i \to \mathcal{A}_i\}$ (i.e. all possible deviations).

**Theorem 4.1** (Complete Classes). *If the reward function class $\mathcal{F}$ and deviation class $\Phi$ are complete and regret equivalence is satisfied (i.e. $\sup_{f \in \mathcal{F}}(\mathcal{R}_\Phi(\sigma, f) - \mathcal{R}_\Phi(\sigma_E, f)) = 0$), then value equivalence is also satisfied:* $\sup_{f \in \mathcal{F}} \max_{i \in [m]}(J_i(\pi_{\sigma_E}, f) - J_i(\pi_\sigma, f)) = 0$. *[Proof]*

Next, we prove that large classes are needed for this implication to hold true.

**Theorem 4.2** (Incomplete Classes). *There exists an MG, an expert policy $\sigma_E$, and a trained policy $\sigma$ such that even though the regret equivalence is satisfied under the true reward function $r$, i.e. $\mathcal{R}_\Phi(\sigma, r) - \mathcal{R}_\Phi(\sigma_E, r) = 0$, the value gap $\max_{i \in [m]}(J_i(\pi_{\sigma_E}, r) - J_i(\pi_\sigma, r)) \neq 0$. [Proof]*

Together, these results tell us that with an expressive enough class of reward functions / deviations, regret equivalence is stronger than value equivalence. We now turn our attention to the converse.

## 4.2 Value Equivalence $\implies$ Regret Equivalence

We now show a surprising result: *value equivalence does not directly imply a low regret gap!* In the worst case, value equivalence fails to provide *any* meaningful guarantees on the regret gap. This reveals a critical distinction between SAIL and MAIL not fully addressed in the prior work.

**Theorem 4.3.** *There exists a Markov Game, an expert policy $\sigma_E$, and a learner policy $\sigma$, such that even occupancy measure of $\pi_\sigma$ exactly matches $\pi_{\sigma_E}$, i.e. $\forall(s, \vec{a}), \rho^{\pi_\sigma}(s, \vec{a}) = \rho^{\pi_{\sigma_E}}(s, \vec{a})$ (i.e. we have value equivalence under all rewards), the regret gap $\mathcal{R}_\Phi(\sigma) - \mathcal{R}_\Phi(\sigma_E) \geq \Omega(H)$. [Proof]*

We leave the details of the proof for this theorem in Appendix E.3. As visualized in Figure 2, both the expert and learner policies only visit the states in the lower path $s_2, s_4, ..., s_{2H-2}$. The trained policy perfectly matches the occupancy measure of the expert by taking identical actions in visited states $s_2, s_4, ..., s_{2H-2}$. However, expert demonstrations lack coverage of state $s_1$ as it is unreachable by executing $\pi_E$. This omission becomes critical when agent 1 deviates from the original policy, making $s_1$ unreachable with high probability. Consequently, the trained policy may perform poorly in $s_1$, in stark contrast to the expert playing a CE under the true reward function. This example highlights the key difference between value equivalence and regret equivalence: the former only depends on states actually visited by the policy, while the latter depends on the counterfactual recommendations the learner would make at unvisited states in response to an agent deviations.

**Remark 4.1.** *As shown in Theorem 4.3, even if the learner has access to infinite samples on the equilibrium path from expert demonstrations, it is possible that the learner remains unaware of the expert's behavior in states unvisited by the expert (but reachable by the deviated agents joint policy). Thus, from an information theoretic perspective, it is impossible for the learner to minimize the regret gap without knowing how the expert would behave on those states. This demonstrates the fundamental difficulty of minimizing the regret gap, and thus, **regret is 'hard' in MAIL**. We therefore need a fundamentally new paradigm of MAIL algorithm to minimize the regret gap.*

---

[1] We prove in Appendix D that the value and regret gaps are equivalent in single-agent IL.

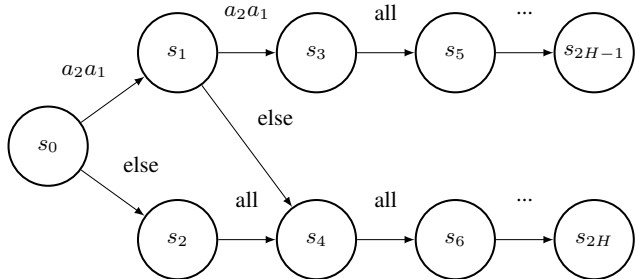

Figure 2: Illustration of an Markov Game that captures why *"regret is hard"*. Here, $\sigma_E(a_1a_1|s_0) = 1$. Observe that $s_1$ is un-visited when all agents obediently follow $\sigma_E$ but is with probability 1 under deviation $\phi_1$ ($\phi_1(s_0, a_1) = \phi_1(s_1, a_1) = a_2$). This means that unless we know what the expert $\sigma_E$ would have recommended counter-factually in $s_1$, we cannot minimize the regret gap.

## 4.3 Low Regret Gap $\implies$ CE, Low Value Gap $\not\implies$ CE

Given the deep connections between regret and correlated equilibrium discussed above, it is perhaps intuitive that if the expert $\sigma_E$ is playing a CE, a low regret gap means the learner is as well.

**Theorem 4.4** (Regret Gap Implies CE). *If the expert policy $\sigma_E$ induces a $\delta_1$-approximate CE, and the learner policy $\sigma$ satisfies $\mathcal{R}_\Phi(\sigma) - \mathcal{R}_\Phi(\sigma_E) \leq \delta_2$, then $\sigma$ induces a $\delta_1 + \delta_2$-approximate CE.* *[Proof]*

Then, by combining our preceding result with Theorem 4.3, it follows that a low value gap does not imply that the learner is playing a CE.

**Corollary 4.5.** *There exists a Markov Game, an expert policy $\sigma_E$, and a learner policy $\sigma$, such that $\sigma_E$ induces a $\delta_1$-approximate CE, and $\sigma$ satisfies $\max_{i \in [m]}(J_i(\pi_{\sigma_E}) - J_i(\pi_\sigma)) = \delta_2$, $\sigma$ induces a $\Omega(H)$-approximate CE.*

Together, these results imply that if we are interested in inducing a CE amongst the agents in the population, the regret gap is a more suitable objective.

## 4.4 Efficient Algorithms for Minimizing the Value Gap

Although we have shown that the value gap is a 'weaker' objective in some sense, in many real-world scenarios, the agents may be non-strategic. In these scenarios, minimizing value gap can be a reasonable learning objective. As we will demonstrate here, the natural multi-agent generalization of single-agent IL algorithms can efficiently minimize the value gap—hence, ***value is 'easy' in MAIL.***

Behavior Cloning (BC) and Inverse Reinforcement Learning (IRL) are two single-agent IL algorithms aimed at minimizing the value gap. By running these algorithms over joint policies, we can apply BC and IRL to the multi-agent setting, which we call Joint Behavior Cloning (J-BC) and Joint Inverse Reinforcement Learning (J-IRL). Doing so results in the same value gap bounds as in the single-agent setting. More details on of J-BC and J-IRL can be found in Appendix B.

**Theorem 4.6** (J-BC Value Gap Upper Bound). *If J-BC returns a policy $\sigma$ that satisfies $\mathbb{E}_{s \sim d^{\pi_{\sigma_E}}}[\ell(\sigma_E(s), \sigma(s))] \leq \epsilon$, then the value gap $\max_{i \in [m]}(J_i(\pi_{\sigma_E}) - J_i(\pi_\sigma)) \leq O(\epsilon H^2)$. [Proof]*

**Theorem 4.7** (J-IRL Value Gap Upper Bound). *If J-IRL outputs a policy $\sigma$ with moment-matching error*

$$\sup_{f \in \mathcal{F}} \mathbb{E}_{\pi_{\sigma_E}} \left[ \sum_{h=1}^{H} f(s_h, \vec{a}_h) \right] - \mathbb{E}_{\pi_\sigma} \left[ \sum_{h=1}^{H} f(s_h, \vec{a}_h) \right] \leq \epsilon H,$$

*then the value gap $\max_{i \in [m]}(J_i(\pi_{\sigma_E}) - J_i(\pi_\sigma)) \leq O(\epsilon H)$. [Proof]*

As argued by Swamy et al. [21], satisfying the conditions for either of the above theorems can be achieved oracle-efficiently via a reduction to no-regret online learning. We now turn our attention to sufficient conditions for there to exist efficient algorithms for minimizing the regret gap.

# 5 Efficient Algorithms for Minimizing the Regret Gap

In our following analysis, we will make a *recoverability* assumption: that a single-step agents deviation could at most cost the expert a fixed constant.

**Assumption 5.1** ($u$-recoverability). *We say that an MG is $u$-recoverable if the expert advantage function is bounded for all deviations, i.e. $\forall s, \vec{a}, h, i, \phi_i, \left| A_{i,h}^{\pi_{\sigma_E}, \phi_i}(s, \vec{a}) \right| \leq u$.*

Intuitively, a small value of $u$ means that we're not in a problem where a single agent can deviate and a joint mistake happens that even the expert couldn't recover from for the rest of the episode. In the worst case, $u$ is $O(H)$. However, we believe that in many cases $u$ is small. For instance, in the route planning example, at some point many cars may miss their turns/intersections/exits, but this can be recovered within a constant time, even when a single driver chooses not to follow its recommendation, rather than $u$ increasing as $H$ increases.

This assumption can be thought of natural multi-agent generalization of the standard recoverability assumption in SAIL [18, 21, 20] which is necessary and sufficient to avoid compounding errors while maintaining computational efficiency. While we define recoverability with respect to the actual reward function for implicitly, one can instead easily define it with respect to the worst-case reward function in a class ($\sup_{f \in \mathcal{F}}$) – *moment* recoverability – as in [21] to avoid the need to know the ground truth set of agent reward functions $r$ to bound $u$.

In Section 4.2, we proved that for general MGs, J-BC and J-IRL don't give any guarantees on the regret gap. Fundamentally, without the ability to observe how the expert would have responded in the counter-factual state induced by a deviation, the learner cannot ensure that they match the expert's regret. We now explore two different sets of assumptions that give us this ability.

## 5.1 Assumption 1: Full Coverage of Expert Demonstrations

In this section, we introduce a coverage assumption on the expert's state distribution $d^{\pi_{\sigma_E}}(s)$ which states that the expert visits every state with a positive probability. We will show that this assumption is sufficient to give a regret gap guarantee. The state coverage assumption is a common theoretical assumption in the analysis of learning in MDPs/MGs [5] and has been explored in SAIL [20].

**Assumption 5.2** ($\beta$-coverage). *There exists a constant $\beta > 0$ such that for the expert's policy $\sigma_E$, it holds that $d^{\pi_{\sigma_E}}(s) \geq \beta$ for all $s$.*

Intuitively, this assumption implies that in the infinite sample limit, there are no states where we are unsure what the expert would recommend. As discussed in Remark 4.1, without the ability to interactively query the expert, a coverage assumption is necessary because we cannot minimize the regret gap without knowing the expert mediator's actions in counter-factual states.

We first show that under Assumption 5.2, J-BC and J-IRL get a (relatively weak) regret gap guarantee.

### 5.1.1 Regret Gaps of J-BC and J-IRL under Full Demonstration Coverage

We begin by analyzing joint behavioral cloning (J-BC).

**Theorem 5.1** (J-BC Regret Gap Upper Bound). *Under Assumption 5.1 and Assumption 5.2, if the J-BC algorithm returns a policy $\sigma$ that satisfies $\mathbb{E}_{s \sim d^{\pi_{\sigma_E}}} [\ell(\sigma_E(s), \sigma(s))] \leq \epsilon$, then*

$$\mathcal{R}_\Phi(\sigma) - \mathcal{R}_\Phi(\sigma_E) \leq O\left(\frac{1}{\beta} \epsilon u H\right).$$

*[Proof]*

We leave the proof in Appendix E.7. It is worth to note that although the dependency of $H$ is linear under our recoverability assumption, we still need to pay for the term $\frac{1}{\beta}$ in our regret gap bound. In general, this term can grow exponentially with the horizon, making this guarantee relatively weak. We can show its tightness by slightly modifying the example in Theorem 4.3 to satisfy the assumptions.

**Theorem 5.2** (J-BC Regret Gap Lower Bound). *There exists a Markov Game, an expert policy $\sigma_E$, and learner policy $\sigma$ such that $\sigma_E$ satisfies Assumption 5.1 and Assumption 5.2, $\sigma$ achieves BC error*

$\mathbb{E}_{s \sim d^{\pi_{\sigma_E}}}[\ell_{\mathsf{TV}}(\sigma_E(s), \sigma(s))] \leq \epsilon$, and

$$\mathcal{R}_\Phi(\sigma) - \mathcal{R}_\Phi(\sigma_E) = \Omega\left(\frac{1}{\beta}\epsilon u H\right).$$

*[Proof]*

We now prove analogous results for joint inverse reinforcement learning (J-IRL).

**Theorem 5.3** (J-IRL Regret Gap Upper Bound). *Under Assumption 5.2 and Assumption 5.1 and with a complete reward function class $\mathcal{F}$, if J-IRL returns a policy $\sigma$ with moment-matching error*

$$\sup_{f \in \mathcal{F}} \mathbb{E}_{\pi_{\sigma_E}}\left[\frac{\sum_{h=1}^{H} f(s_h, \vec{a}_h)}{H}\right] - \mathbb{E}_{\pi_\sigma}\left[\frac{\sum_{h=1}^{H} f(s_h, \vec{a}_h)}{H}\right] \leq \epsilon,$$

*then $\mathcal{R}_\Phi(\sigma) - \mathcal{R}_\Phi(\sigma_E) \leq O\left(\frac{1}{\beta}\epsilon u H\right)$. [Proof]*

There are two interesting features of this theorem. The first is that we needed to assume that the reward function class is complete – otherwise, a small value gap can still translate to a large regret gap. The second is that the upper-bound for J-IRL matches that for J-BC, which is in stark contrast to the single-agent setting, where IRL enjoys linear-in-$H$ guarantees with respect to the value gap [21]. We now show this is not an artifact of our analysis by providing a matching lower bound.

**Corollary 5.4** (J-IRL Regret Gap Lower Bound). *There exists a Markov Game, an expert policy $\sigma_E$, and a policy $\sigma$ such that $\sigma_E$ satisfies Assumption 5.1 and Assumption 5.2, the trained policy $\sigma$ gets moment-matching error*

$$\sup_{f \in \mathcal{F}} \mathbb{E}_{\pi_{\sigma_E}}\left[\frac{\sum_{h=1}^{H} f(s_h, \vec{a}_h)}{H}\right] - \mathbb{E}_{\pi_\sigma}\left[\frac{\sum_{h=1}^{H} f(s_h, \vec{a}_h)}{H}\right] \leq \epsilon,$$

*and $\mathcal{R}_\Phi(\sigma) - \mathcal{R}_\Phi(\sigma_E) = \Omega\left(\frac{1}{\beta}\epsilon u H\right)$. [Proof]*

This result implies another fundamental distinction between SAIL and MAIL: ***in contrast to the value gap, interactive training alone is not sufficient to effectively minimize the regret gap.***

### 5.1.2 `MALICE`: Multi-agent Aggregation of Losses to Imitate Cached Experts

Observe that the upper bounds for both J-BC and J-IRL include a dependence on the inverse of the coverage coefficient $\frac{1}{\beta}$, which can be rather large for problems with long horizons or large action spaces. We now present an efficient algorithm that is able to avoid this dependence by extending the ALICE algorithm [20] to the multi-agent setting. ALICE is an interactive algorithm that, at each round, uses importance sampling to re-weight the behavior cloning (BC) loss based on the density ratio between the current learner policy and that of the expert. Accordingly, ALICE requires a full demonstration coverage assumption to ensure that these importance weights are finite. ALICE uses a no-regret algorithm to learn a policy that minimizes reweighed on-policy error, which guarantees a linear-in-$H$ bound on the value gap under a recoverability assumption [20].

In Algorithm 1, we describe Multi-agent ALICE (`MALICE`), where adapt ALICE to the multi-agent setting (i.e. minimizing the regret gap). Specifically, we modify the ALICE loss function to include a maximum over all deviations. This gives us

$$\ell_{\mathsf{MALICE}}(\sigma, D_E, \hat{\sigma}) = \max_{i \in [m]} \max_{\phi_i \in \Phi_i} \mathbb{E}_{s \sim d^{\pi_{\sigma_E}}}\left[\frac{d^{\pi_{\hat{\sigma}, \phi_i}}(s)}{d^{\pi_{\sigma_E}}(s)}\ell(\sigma_E(s), \sigma(s))\right]. \tag{5}$$

Since $\mathbb{E}_{s \sim d^{\pi_E}}\left[\frac{d^{\pi_{\hat{\sigma}, \phi_i}}(s)}{d^{\pi_{\sigma_E}}(s)}\ell(\sigma_E(s), \sigma(s))\right]$ is a convex loss function, and the maximum of convex functions is still a convex function, we know that $\ell_{\mathsf{MALICE}}(\sigma, D_E, \hat{\sigma})$ is a valid convex loss function with scales in $[0, 1]$. As a result, we can run an (arbitrary) no-regret online convex optimization (OCO) algorithm to efficiently optimize it, giving us an ***efficient reduction from regret gap minimization to no-regret online convex optimization under demonstration coverage***.

We now provide regret gap guarantees on the policy returned by `MALICE`.

---

**Algorithm 1** `MALICE` (Multi-agent Aggregation of Losses to Imitate Cached Experts)

---
1: **Input:** Expert demonstrations $D_E$.
2: Initialize $\sigma^{(1)} \in \Sigma$.
3: **for** $n = 1$ **to** $N$ **do**
4:    **for** $i = 1$ **to** $m$ **do**
5:       **for** $\phi_i \in \Phi_i$ **do**
6:          Sample states from $s_t \sim d^{\pi_{\sigma,\phi_i}^{(n)}}$.
7:       **end for**
8:    **end for**
9:    Construct loss function $\ell^{(n)}(\sigma) = \ell_{\text{MALICE}}(\sigma, D_E, \sigma^{(n)})$.
10:    `// Run arbitrary no-regret OCO algorithm on sequence of losses, e.g.  FT(R)L:`
11:    $\sigma^{(n+1)} \leftarrow \arg\min_{\sigma \in \Sigma} \sum_{j=1}^{n} \ell^{(n)}(\sigma)$
12: **end for**
13: **Return** Best of $\sigma^{(1:N)}$ on validation data.

---

**Theorem 5.5** (`MALICE` Regret Gap Upper Bound). *Let $\sigma$ be a policy such that $\ell_{\text{MALICE}}(\sigma, D_E, \sigma) \leq \epsilon$. Under Assumption 5.1 and Assumption 5.2, we have*

$$\mathcal{R}_\Phi(\sigma) - \mathcal{R}_\Phi(\sigma_E) \leq O(\epsilon u H).$$

*[Proof]*

As promised, observe that adapting the importance sampling technique of Spencer et al. [20] to the multi-agent setting allows us to efficiently minimize the regret gap while avoiding an upper bound that depends on the coverage coefficient of the expert demonstrations.

We now show that the bound in Theorem 5.5 is tight by constructing a matching lower bound.

**Theorem 5.6** (`MALICE` Regret Gap Lower Bound). *There exists a Markov Game, an expert policy $\sigma_E$ that satisfies Assumption 5.1, and a trained policy $\sigma$ that gets error $\ell_{\text{TV,MALICE}}(\sigma, D_E, \sigma) \leq \epsilon$, and*

$$\mathcal{R}_\Phi(\sigma) - \mathcal{R}_\Phi(\sigma_E) = \Omega\left(\epsilon u H\right).$$

*[Proof]*

We now turn our attention to an alternate assumption and the corresponding regret gap algorithm.

## 5.2 Assumption 2: Access to a Queryable Expert

For many problems, full coverage of expert demonstrations is not a reasonable assumption. Thus, we explore another natural assumption that allows us to observe expert recommendations at counterfactual states: access to a queryable expert. In their classic DAgger algorithm, Ross et al. [18] showed that access to a queryable expert allows one to eliminate the covariate shift that results from the difference between expert and learner induced state distributions. When we transition to the multi-agent setting, we can again use access to a queryable expert to handle yet another source of covariate shift: potential strategic deviations by agents in the population that push the learner outside of the support of the expert. We refer to our multi-agent extension of DAgger as `BLADES`.

In each iteration of `BLADES`, we request the expert to provide recommendations under all possible agent deviations, before training on the aggregated data. More formally, we minimize the following sequence of loss functions:

$$\ell_{\text{BLADES}}(\sigma, \hat{\sigma}) = \max_{i \in [m]} \max_{\phi_i \in \Phi_i} \mathbb{E}_{s \sim d^{\pi_{\hat{\sigma},\phi_i}}}[\ell(\sigma_E(s), \sigma(s))]. \tag{6}$$

Similar to `MALICE`, we know that the loss $\ell_{\text{BLADES}}$ is also a valid convex loss function, and thus we can use a no-regret algorithm to efficiently minimize it. This gives us an ***efficient reduction from regret gap minimization to no-regret online convex optimization with access to a queryable expert.*** We now derive and upper and lower bounds on the regret gap of a policy returned by `BLADES`.

**Theorem 5.7** (`BLADES` Regret Gap Upper Bound). *Under Assumption 5.1, if a policy $\sigma$ satisfies $\ell_{\text{BLADES}}(\sigma, \sigma) \leq \epsilon$, then*

$$\mathcal{R}_\Phi(\sigma) - \mathcal{R}_\Phi(\sigma_E) \leq O(\epsilon u H).$$

*[Proof]*

**Algorithm 2** BLADES (Bend Learner, Aggregate Datasets of Expert Suggestions)

1: **Input:** Expert demonstrations $D_E$.
2: Initialize learner $\sigma^{(1)} = \arg\min_\sigma \mathbb{E}_{s \sim D_E} \ell(\sigma_E(s), \sigma(s))$.
3: **for** $n = 1$ **to** $N$ **do**
4:     **for** $i = 1$ **to** $m$ **do**
5:         **for** $\phi_i \in \Phi_i$ **do**
6:             Sample trajectories from $\pi_{\sigma,\phi_i}^{(n)}$.
7:             Query expert for action recommendations to construct dataset $D_{\phi_i}^{(n)} = \{(s, \sigma_E(s))\}$.
8:         **end for**
9:     **end for**
10:     Construct loss function $\ell^{(n)}(\sigma) = \ell_{\text{BLADES}}(\sigma, \sigma^{(n)})$.
11:     // Run arbitrary no-regret OCO algorithm on sequence of losses, e.g. FT(R)L:
12:     $\sigma^{(n+1)} \leftarrow \arg\min_{\sigma \in \Sigma} \sum_{j=1}^n \ell^{(n)}(\sigma)$.
13: **end for**
14: **Return** Best of $\sigma^{(1:N)}$ on validation data.

**Theorem 5.8** (BLADES Regret Gap Lower Bound). *There exists a Markov Game, an expert policy $\sigma_E$, and a trained policy $\sigma$ such that $\sigma_E$ satisfies Assumption 5.1, $\sigma$ achieves error $\ell_{\text{TV,BLADES}}(\sigma, \sigma) \leq \epsilon$, and*

$$\mathcal{R}_\Phi(\sigma) - \mathcal{R}_\Phi(\sigma_E) = \Omega\left(\epsilon u H\right).$$

*[Proof]*

In short, under either a demonstration coverage assumption or with access to a queryable expert, we are able to efficiently minimize the regret gap on a recoverable MAIL problem.

# 6 Conclusion

Our work focuses on the core question of what fundamentally distinguishes multi-agent IL problems from single-agent ones. In short, our answer is that on problems with strategic agents that are not mere puppets, we need to deal with another source of distribution shift: deviations by agents in the population. This new source of distribution shift cannot be efficiently controlled with environment interaction (i.e. inverse RL). Instead, we need to be able to estimate how the expert would act in counter-factual states. Based on this core insight, we derive two reductions that are able to minimize the regret gap under a coverage or queryable expert assumption. We leave the development and implementation of practical approximations of our idealized algorithms to future work.

# 7 Acknowledgements

We thank Drew Bagnell and Brian Ziebart for their incredible patience and detailed answers to a somewhat absurdly large number of questions about their prior work. We thank Simon Shaolei Du for a discussion on our coverage assumptions and Noah Golowich for references to relevant lower bounds in the MARL literature. We also thank Sanjiban Choudhury and Wen Sun for comments on our draft. ZSW, GS, and Jingwu Tang are supported in part by an Air Force STTR grant and the NSF Award #1763786. ZSW is supported in part by the NSF FAI Award #1939606, a Google Faculty Research Award, a J.P. Morgan Faculty Award, a Facebook Research Award, an Okawa Foundation Research Grant, and a Mozilla Research Grant. FF is supported in part by NSF grant IIS-2046640 (CAREER) and the Sloan Research Fellowship. GS is supported by his family and friends.

# 8 Contribution Statements

1. **JT** uncovered the distinction between the regret and value gaps, proved all of the core results in the paper, and drafted the initial version of the paper.
2. **GS** initially proposed the project, came up with the sufficient conditions and associated algorithms for minimizing the regret gap, and wrote most of the final version of the paper.
3. **FF** and **ZSW** advised the project.

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

## A  Broader Impacts

As the algorithms we proposed are theoretical, we do not foresee any direct societal concerns resulting from this work. However, these theoretical algorithms can serve as a foundation for developing practical algorithms or provide guidance for designing practical algorithms in MAIL, which could be applied to real world problems in the future.

## B  Extending Single-Agent IL Algorithms to Minimize the Value Gap

### B.1  Multi-Agent Joint Behavior Cloning

Behavioral Cloning (BC, Pomerleau [15]) treats the problem of imitation learning as supervised learning and performs maximum likelihood estimation with expert states as inputs and expert actions as labels. Unfortunately, as first analyzed by Ross and Bagnell [17], the covariate shift between the training (expert states) and test (learner states) distributions can lead to *compounding errors* – i.e. a value gap that increases quadratically as a function of the horizon $H$. We note that this is not an artifact of the particular objective used in BC – as argued by Swamy et al. [21], the same can be said for *any* offline imitation learning algorithm. J-BC extends BC to a multi-agent setting by learning a map from the state space $\mathcal{S}$ to the joint action space $\mathcal{A}$. By adapting the analysis of Ross and Bagnell [17] and Swamy et al. [21] to the multi-agent setting, we establish a similar compounding error result for multi-agent behavior cloning in Theorem 4.6. There exists an example of MDP/MG that matches this bound, which shows that the bound is tight [21].

### B.2  Multi-Agent Inverse Reinforcement Learning

A popular family of online techniques for imitation learning is *inverse reinforcement learning* (IRL). Intuitively, IRL can be thought of as being similar to a GAN [9] but in the space of trajectories: the generator is the learner's policy coupled with a world model to actually give us trajectories, while the discriminator is trained between expert and learner trajectories and is used as a reward function for policy updates. More formally, IRL can be viewed as a two-player zero-sum game between a *reward player* and a *policy player* [21]. In each round, the reward player picks a reward function from $\mathcal{F}$ that maximizes the value gap between $\sigma_E$ and $\sigma$, while the policy player uses a reinforcement learning algorithm to learn a new policy in $\Sigma$ that maximizes the performance under this reward function.

Intuitively, as the learner can see policy rollouts during training procedure, they cannot be "surprised" by where their policy ends up at test time, removing the covariate shift issue that lies at the heart of compounding errors. More formally, Swamy et al. [21] proved that value gap for single-agent IRL algorithm is $O(\epsilon H)$. We now generalize this result to the multi-agent setting. Accordingly, our policy class $\Sigma$ becomes one of joint policies. We use a reward function class $\mathcal{F}$ that is identical for all agents (i.e. we assume the the game is common payoff). Then, by following the proof in Swamy et al. [21], we prove a $O(\epsilon H)$ value gap bound for multi-agent IRL algorithm in Theorem 4.7.

---

**Algorithm 3** J-IRL

1: **Input:** expert demonstration $D_E$, Policy class $\Sigma$, Reward class $\mathcal{F}$
2: Set $\sigma^{(1)} \in \Sigma$
3: **for** $n = 1$ **to** $N$ **do**
4:   $f^{(n)} \leftarrow \arg\max J(\pi_{\sigma_E}, f) - J(\text{Unif}(\pi_{\sigma^{(1:n)}}), f) + R(f)$
     // Treat it as a single-agent RL problem over joint action space under reward
     function $f^{(n)}$
5:   $\sigma^{(n+1)} \leftarrow \text{MaxEntRL}(r = f^{(n)})$
6: **end for**
7: **Return** best $\sigma^{(n)}$ on validation

---

## C  Useful Lemmas

We introduce a lemma which will be very useful in the analysis under the recoverability assumption. It is used in the analysis in the single-agent DAgger [18] and ALICE [20], and we will also use

it in the analysis for MAIL. It shows that if the policy achieves small on-policy error, then, with recoverability assumption, the value gap is linear over $H$.

**Lemma C.1.** *[Ross et al. [18]] For agent joint policy $\pi_1$ and $\pi_2$, if the advantage of $\pi_1$ is bounded under the true reward function $\forall i, h, s, \vec{a}, |A_{i,h}^{\pi_1}(s, \vec{a})| \leq u$, and $\pi_2$ get on-policy error $\mathbb{E}_{s \sim d^{\pi_2}}[\ell(\pi_1(s), \pi_2(s))] \leq \epsilon$, then $|J_i(\pi_1) - J_i(\pi_2)| \leq \epsilon u H, \forall i \in [m]$.*

*Proof.* Via the performance difference lemma, $\forall i \in [m]$, we have

$$
\begin{aligned}
|J_i(\pi_1) - J_i(\pi_2)| &= \left| \sum_{h=1}^{H} \mathbb{E}_{s \sim d_h^{\pi_2}}[A_{i,h}^{\pi_1}(s, \pi(s))] \right| \\
&\leq u H \mathbb{E}_{s \sim d^{\pi_2}}[\ell(\pi_1(s), \pi_2(s))] \\
&\leq \epsilon u H
\end{aligned}
\tag{7}
$$

$\square$

For our analysis of MALICE and BLADES, we will let $\pi_1$ be any deviated expert policy $\pi_{\sigma_E, \phi_i}$ and $\pi_2$ be the deviated trained policy $\pi_{\sigma, \phi_i}$ under the same deviation.

## D   Equivalence of Regret Gap and Value Gap in Single-Agent IL

For single-agent IL we prove that the regret gap and the value gap are equivalent.

**Theorem D.1** (Equivalence in Single-Agent IL). *For single-agent MDP, regret gap and value gap are equivalent to each other*

$$
J(\pi_{\sigma_E}) - J(\pi_\sigma) = \mathcal{R}_\Phi(\sigma) - \mathcal{R}_\Phi(\sigma_E)
$$

*Proof.* For single-agent MDP, we ignore the index $i$ in the following proof. A strategy deviation in single-agent MDP is equivalent to taking another policy, because there are no other agents affecting the dynamics of the agent. We have

$$
\mathcal{R}_\Phi(\sigma) = \max_{\phi \in \Phi}(J(\pi_{\sigma, \phi}) - J(\pi_\sigma)) = J(\pi^*) - J(\pi_\sigma)
$$

where $\pi^*$ is the optimal policy under the true reward function. Similarly, we have

$$
\mathcal{R}_\Phi(\sigma_E) = J(\pi^*) - J(\pi_{\sigma_E})
$$

Therefore,

$$
\mathcal{R}_\Phi(\sigma) - \mathcal{R}_\Phi(\sigma_E) = (J(\pi^*) - J(\pi_\sigma)) - (J(\pi^*) - J(\pi_{\sigma_E})) = J(\pi_{\sigma_E}) - J(\pi_\sigma)
$$

$\square$

In single-agent MDPs, the dynamics are fixed because no other agents affect the agent's dynamics, and therefore, the regret gap is equivalent to the value gap.

# E   Proofs

**Contents**

## E.1   Proof of Theorem 4.1

*Proof.* We prove the lemma by showing that the occupancy measures of $\pi_\sigma$ and $\pi_{\sigma_E}$ exactly match, i.e. $\rho^{\pi_\sigma}(s, \vec{a}) = \rho^{\pi_{\sigma_E}}(s, \vec{a})$ for every $(s, \vec{a})$. Consider a cooperative reward function $f_{s', \vec{a}'} = -\mathbf{1}(s = s', \vec{a} = \vec{a}')$.

Under $f_{s, \vec{a}}$, we have $J(\pi_\sigma) = -H\rho^{\pi_\sigma}(s, \vec{a}), J(\pi_{\sigma_E}) = -H\rho^{\pi_{\sigma_E}}(s, \vec{a})$. The maximum value performance the expert/learner can get after deviation is $0$ because the reward function is non-positive. ($0$ can be achieved by simply not taking $\vec{a}$ on $s$).

Therefore $\mathcal{R}_\Phi(\sigma) = 0 - (-H\rho^{\pi_\sigma}(s, \vec{a})) = H\rho^{\pi_\sigma}(s, \vec{a})$, $\mathcal{R}_\Phi(\sigma_E) = 0 - (-H\rho^{\pi_{\sigma_E}}(s, \vec{a})) = H\rho^{\pi_{\sigma_E}}(s, \vec{a})$.

Since $\mathcal{R}_\Phi(\sigma) - \mathcal{R}_\Phi(\sigma_E) = 0$, we know that $\rho^{\pi_\sigma}(s, \vec{a}) = \rho^{\pi_{\sigma_E}}(s, \vec{a})$. This implies that the occupancy measures of two policies exactly match. As a result,

$$\sup_{f \in \mathcal{F}} \max_{i \in [m]} (J_i(\pi_{\sigma_E}, f) - J_i(\pi_\sigma, f)) = 0$$

$\square$

## E.2   Proof of Theorem 4.2

*Proof.* We can construct an example in normal form games, in which there are mulitple CEs with different pay-offs. We can let the $\sigma_E$ plays CE 1 and $\sigma$ plays CE 2. Therefore, although the regret gap $\mathcal{R}_\Phi(\sigma) - \mathcal{R}_\Phi(\sigma_E) = 0$, the value gap $\max_{i \in [m]} (J_i(\pi_{\sigma_E}) - J_i(\pi_\sigma)) \neq 0$. The NFG in Figure 5 is an example, where $(a_1, a_1)$ and $(a_2, a_2)$ are two CEs with different values.   $\square$

## E.3   Proof of Theorem 4.3

*Proof.* We prove the theorem by constructing such a Markov Game and policies that can get $\Omega(H)$ regret gap. For simplicity, we construct a two-player cooperative game where the reward is identical for all agents. Agents can not visit the same state at different time steps. These allow us to omit the index $i$ in the reward function in the proof. The notation $a_i a_j$ is used to represent the action pair $(a_i, a_j)$.

The transition dynamics are illustrated in Figure 2, and the rewards are action free. The reward function $r(s_3) = r(s_5) = ... = r(s_{2H-3}) = 1$, with all other states yielding a reward of $0$. Each agent has an action space $\mathcal{A}_i = \{a_1, a_2, a_3\}$.

The expert policy $\sigma_E$ satisfies $\sigma_E(a_1a_1|s_0) = 1.\sigma_E(a_3a_3|s_1) = 1$. Action on all other states don't matter because the transition and the reward would be the same. The trained policy $\sigma$ satisfies $\sigma(a_1a_1|s_0) = 1, \sigma(a_1a_1|s_1) = 1$, and plays the same as the expert in all other states.

It is not hard to verify that $\sigma_E$ plays a CE under this reward function, which means

$$\mathcal{R}_\Phi(\sigma_E) = 0$$

The worst deviation for $\sigma$ is to deviate action of agent 1 from playing $a_1$ to $a_2$ on both $s_0$ and $s_1$. We get

$$\mathcal{R}_\Phi(\sigma) = H - 2$$

Therefore, the regret gap $\mathcal{R}_\Phi(\sigma) - \mathcal{R}_\Phi(\sigma_E) = H - 2 = \Omega(H)$ $\qquad \square$

### E.4 Proof of Theorem 4.4

*Proof.* From the definition of CE, we know $\mathcal{R}_\Phi(\sigma_E) \leq \delta_1$. Therefore,

$$\mathcal{R}_\Phi(\sigma) = \mathcal{R}_\Phi(\sigma_E) + (\mathcal{R}_\Phi(\sigma) - \mathcal{R}_\Phi(\sigma_E)) \leq \delta_1 + \delta_2 \qquad (8)$$

Thus, we know that $\sigma$ induces a $\delta_1 + \delta_2$-approximate CE. $\qquad \square$

### E.5 Proof of Theorem 4.6

*Proof.* For any $i$, we can view multi-agent problem as a single agent MDP over the joint action space under reward function $r_i$. Following the proof in Ross and Bagnell [17], Swamy et al. [21], we can prove $J_i(\pi_{\sigma_E}) - J_i(\pi_\sigma) \leq O(\epsilon H^2)$. Therefore, $\max_{i \in [m]}(J_i(\pi_{\sigma_E}) - J_i(\pi_\sigma)) \leq O(\epsilon H^2)$. $\qquad \square$

### E.6 Proof of Theorem 4.7

*Proof.* For any $i$,

$$J_i(\pi_{\sigma_E}) - J_i(\pi_\sigma) \leq \sup_{f \in \mathcal{F}} \mathbb{E}_{\xi \sim \pi_{\sigma_E}} \left[ \sum_{h=1}^{H} f(s_h, \vec{a}_h) \right] - \mathbb{E}_{\xi \sim \pi_\sigma} \left[ \sum_{h=1}^{H} f(s_h, \vec{a}_h) \right] \leq \epsilon H$$

Therefore, $\max_{i \in [m]}(J_i(\pi_{\sigma_E}) - J_i(\pi_\sigma)) \leq O(\epsilon H)$. $\qquad \square$

### E.7 Proof of Theorem 5.1

*Proof.* With Assumption 5.2, we know that

$$\mathbb{E}_{s \sim d^{\pi_\sigma}} [\ell(\sigma_E(s), \sigma(s))] \leq \frac{1}{\beta} \mathbb{E}_{s \sim d^{\pi_{\sigma_E}}} [\ell(\sigma_E(s), \sigma(s))] \leq \frac{\epsilon}{\beta}$$

By Lemma C.1, we get

$$J_i(\pi_{\sigma_E}) - J_i(\pi_\sigma) \leq O\left( \frac{1}{\beta} \epsilon u H \right)$$

For any deviation $\phi_i$,

$$\mathbb{E}_{s \sim d^{\pi_\sigma, \phi_i}} [\mathsf{TV}(\pi_{\sigma_E, \phi_i}(s), \pi_{\sigma, \phi_i}(s))] \leq \mathbb{E}_{s \sim d^{\pi_\sigma, \phi_i}} [\mathsf{TV}(\pi_{\sigma_E}(s), \pi_\sigma(s))] \leq \frac{1}{\beta} \mathbb{E}_{s \sim d^{\pi_{\sigma_E}}} [\mathsf{TV}(\sigma_E(s), \sigma(s))] \leq \frac{\epsilon}{\beta}$$

By Lemma C.1, we get

$$J_i(\pi_{\sigma, \phi_i}) - J_i(\pi_{\sigma_E, \phi_i}) \leq O\left( \frac{1}{\beta} \epsilon u H \right)$$

Therefore,

$$J_i(\pi_{\sigma, \phi_i}) - J_i(\pi_\sigma) = (J_i(\pi_{\sigma, \phi_i}) - J_i(\pi_{\sigma_E, \phi_i})) + (J_i(\pi_{\sigma_E, \phi_i})) - J_i(\pi_{\sigma_E})) + (J_i(\pi_{\sigma_E}) - J_i(\pi_\sigma))$$

$$\leq J_i(\pi_{\sigma_E, \phi_i}) - J_i(\pi_{\sigma_E}) + O\left( \frac{1}{\beta} \epsilon u H \right)$$

$$(9)$$

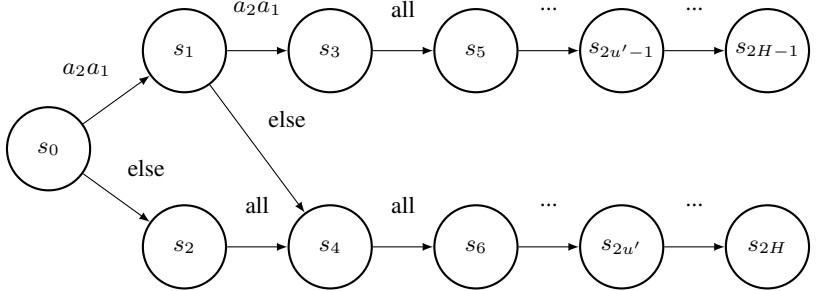

Figure 3: Example of $\Omega(\frac{1}{\beta}\epsilon u H)$ regret gap for J-BC and J-IRL

Taking the maximum over $i, \phi_i$, we get

$$\mathcal{R}_\Phi(\sigma) - \mathcal{R}_\Phi(\sigma_E) \leq O\left(\frac{1}{\beta}\epsilon u H\right)$$

$\square$

### E.8 Proof of Theorem 5.2

*Proof.* We prove the theorem by constructing such a Markov Game policies that can get $\Omega(\frac{1}{\beta}\epsilon u H)$ regret gap. We consider the two-player cooperative game similar to the example in Theorem 4.3. What we need to do is to slightly modify the MG and the policy to satisfy Assumption 5.1 and Assumption 5.2. The rewards are action free. Let $u' = \lfloor u \rfloor$, the reward function $r(s_3) = r(s_5) = \ldots = r(s_{2u'-3}) = 1$, with all other states yielding a reward of 0. The transition of the MG is shown in Figure 3. We know that the value is between $[0, u']$ for any policy, which means Assumption 5.1 is satisfied.

Let $\sigma_E$ be the policy that $\sigma_E(a_1 a_1|s_0) = 1 - 2\beta, \sigma_E(a_2 a_1|s_0) = 2\beta, \sigma_E(a_2 a_1|s1) = \frac{1}{2}, \sigma_E(a_3 a_3|s_1) = \frac{1}{2}$. Action at all other states doesn't matter because the transition and the reward would be the same. $\sigma_E$ satisfies Assumption 5.2.

Let trained policy $\sigma$ be the policy that $\sigma(a_1 a_1|s_0) = 1 - 2\beta, \sigma(a_2 a_1|s_0) = 2\beta, \sigma(a_2 a_1|s1) = \frac{1}{2}, \sigma(a_1 a_1|s1) = \frac{\epsilon H}{2\beta}, \sigma(a_3 a_3|s_1) = \frac{1}{2} - \frac{\epsilon H}{2\beta}$. $\sigma$ and $\sigma_E$ only differs at $s_1$.

Behavior cloning error of $\sigma$ satisfies

$$\mathbb{E}_{s \sim d^{\pi_{\sigma_E}}}[\ell_{\mathsf{TV}}(\sigma_E(s), \sigma(s))] \leq 2\beta \cdot \frac{\epsilon H}{2\beta} \cdot \frac{1}{H} = \epsilon$$

It is not hard to verify, the worst deviation for $\pi_{\sigma_E}$ is to deviate action of agent 1 at $s_0$ from playing $a_1$ to $a_2$, and thus

$$\mathcal{R}_\Phi(\sigma_E) = \frac{1}{2}(1 - 2\beta)(u' - 2)$$

The worst deviation of $\pi_\sigma$ is to deviate action of agent 1 from playing $a_1$ to $a_2$ at $s_0$ and $s_1$.

$$\mathcal{R}_\Phi(\sigma) = \frac{1}{2}(1 - 2\beta)(u' - 2) + \frac{\epsilon H}{2\beta}(u' - 2)$$

Therefore, the regret gap $\mathcal{R}_\Phi(\sigma) - \mathcal{R}_\Phi(\sigma_E) = \frac{\epsilon H}{2\beta}(u' - 2) = \Omega(\frac{1}{\beta}\epsilon u H)$. $\square$

### E.9 Proof of Theorem 5.3

*Proof.* We prove it by showing that under complete reward function class $\mathcal{F}$, low IRL error will imply low BC error, and then apply Theorem 5.1.

When $\mathcal{F} = [-1,1]^{|\mathcal{S}||\mathcal{A}|}$,

$$\sup_{f \in \mathcal{F}} \mathbb{E}_{\pi_{\sigma_E}} \left[ \frac{\sum_{h=1}^{H} f(s_h, \vec{a}_h)}{H} \right] - \mathbb{E}_{\pi_\sigma} \left[ \frac{\sum_{h=1}^{H} f(s_h, \vec{a}_h)}{H} \right]$$

$$= \sup_{f \in \mathcal{F}} \sum_{s, \vec{a}} [\rho^{\pi_{\sigma_E}}(s, \vec{a}) - \rho^{\pi_\sigma}(s, \vec{a})] f(s, \vec{a}) \tag{10}$$

$$= \sum_{s, \vec{a}} |\rho^{\pi_{\sigma_E}}(s, \vec{a}) - \rho^{\pi_\sigma}(s, \vec{a})|$$

Therefore, we have $\sum_{s, \vec{a}} |\rho^{\pi_{\sigma_E}}(s, \vec{a}) - \rho^{\pi_\sigma}(s, \vec{a})| \le \epsilon$.

$$\sum_{s, \vec{a}} |\rho^{\pi_{\sigma_E}}(s, \vec{a}) - \rho^{\pi_\sigma}(s, \vec{a})|$$

$$= \sum_{s, \vec{a}} |d^{\pi_{\sigma_E}}(s) \sigma_E(\vec{a}|s) - d^{\pi_\sigma}(s) \sigma(\vec{a}|s)|$$

$$= \sum_{s, \vec{a}} |d^{\pi_{\sigma_E}}(s) \sigma_E(\vec{a}|s) - d^{\pi_{\sigma_E}}(s) \sigma(\vec{a}|s) + d^{\pi_{\sigma_E}}(s) \sigma(\vec{a}|s) - d^{\pi_\sigma}(s) \sigma(\vec{a}|s)|$$

$$\ge \sum_{s, \vec{a}} (|d^{\pi_{\sigma_E}}(s) \sigma_E(\vec{a}|s) - d^{\pi_{\sigma_E}}(s) \sigma(\vec{a}|s)| - |d^{\pi_{\sigma_E}}(s) \sigma(\vec{a}|s) - d^{\pi_\sigma}(s) \sigma(\vec{a}|s)|) \tag{11}$$

$$= \mathbb{E}_{s \sim d^{\pi_{\sigma_E}}} [\mathsf{TV}(\sigma_E(s), \sigma(s))] - \sum_s |d^{\pi_{\sigma_E}}(s) - d^{\pi_\sigma}(s)|$$

$$= \mathbb{E}_{s \sim d^{\pi_{\sigma_E}}} [\mathsf{TV}(\sigma_E(s), \sigma(s))] - \sum_s \left| \sum_a [\rho^{\pi_{\sigma_E}}(s, \vec{a}) - \rho^{\pi_\sigma}(s, \vec{a})] \right|$$

$$\ge \mathbb{E}_{s \sim d^{\pi_{\sigma_E}}} [\mathsf{TV}(\sigma_E(s), \sigma(s))] - \sum_{s, a} |\rho^{\pi_{\sigma_E}}(s, \vec{a}) - \rho^{\pi_\sigma}(s, \vec{a})|$$

$$\ge \mathbb{E}_{s \sim d^{\pi_{\sigma_E}}} [\mathsf{TV}(\sigma_E(s), \sigma(s))] - \epsilon$$

Therefore, we get

$$\mathbb{E}_{s \sim d^{\pi_{\sigma_E}}} [\mathsf{TV}(\sigma_E(s), \sigma(s))] \le 2\epsilon$$

Directly applying Theorem 5.1, we get $\mathcal{R}_\Phi(\sigma) - \mathcal{R}_\Phi(\sigma_E) \le O\left( \frac{1}{\beta} \epsilon u H \right)$. $\qquad \square$

### E.10 Proof of Corollary 5.4

*Proof.* Consider the same example in proof of Theorem 5.2 with parameter $\epsilon'$. In the example, the only difference between the occupancy measures of two policies are $\rho(s, \vec{a})$ at state $s_1$. Therefore,

$$\sup_{f \in \mathcal{F}} \mathbb{E}_{\pi_{\sigma_E}} \left[ \frac{\sum_{h=1}^{H} f(s_h, \vec{a}_h)}{H} \right] - \mathbb{E}_{\pi_\sigma} \left[ \frac{\sum_{h=1}^{H} f(s_h, \vec{a}_h)}{H} \right]$$

$$= \sup_{f \in \mathcal{F}} \sum_{s, \vec{a}} [\rho^{\pi_{\sigma_E}}(s, \vec{a}) - \rho^{\pi_\sigma}(s, \vec{a})] f(s, \vec{a})$$

$$\le \sum_{s, \vec{a}} |\rho^{\pi_{\sigma_E}}(s, \vec{a}) - \rho^{\pi_\sigma}(s, \vec{a})| \tag{12}$$

$$\le |\rho^{\pi_{\sigma_E}}(s_1, a_3 a_3) - \rho^{\pi_\sigma}(s_1, a_3 a_3)| + |\rho^{\pi_{\sigma_E}}(s_1, a_1 a_1) - \rho^{\pi_\sigma}(s_1, a_1 a_1)|$$

$$= \frac{1}{H} \left( 2\beta \cdot \frac{\epsilon' H}{2\beta} \cdot 2 \right) = 2\epsilon'$$

Let $\epsilon' = \frac{1}{2} \epsilon$. Then the regret gap $\mathcal{R}_\Phi(\sigma) - \mathcal{R}_\Phi(\sigma_E) = \frac{\epsilon H}{4\beta}(u' - 2) = \Omega(\frac{1}{\beta} \epsilon u H)$. $\qquad \square$

### E.11 Proof of Theorem 5.5

*Proof.* From the definition of $\ell_{\text{MALICE}}$, we know

$$\ell_{\text{MALICE}}(\sigma, D_E, \sigma) = \max_{i \in [m]} \max_{\phi_i} \mathbb{E}_{s \sim d^{\pi_{\sigma_E}}} \left[ \frac{d^{\pi_{\sigma,\phi_i}}}{d^{\pi_{\sigma_E}}} \ell(\pi_E(s), \pi(s)) \right]$$

$$\geq \max_{i \in [m]} \max_{\phi_i} \mathbb{E}_{s \sim d^{\pi_{\sigma_E}}} \left[ \frac{d^{\pi_{\sigma,\phi_i}}}{d^{\pi_{\sigma_E}}} \ell(\pi_{E\phi_i}(s), \pi_{\phi_i}(s)) \right] \quad (13)$$

$$\geq \max_{i \in [m]} \max_{\phi_i} \mathbb{E}_{s \sim d^{\pi_{\sigma,\phi_i}}} \left[ \ell(\pi_{E\phi_i}(s), \pi_{\phi_i}(s)) \right]$$

From Lemma C.1, we know that for all $i, \phi_i$, we have

$$J_i(\pi_{\sigma,\phi_i}) - J_i(\pi_{\sigma_E,\phi_i}) \leq O(\epsilon u H)$$

And

$$J_i(\pi_{\sigma_E}) - J_i(\pi_\sigma) \leq O(\epsilon u H)$$

Therefore, we get

$$J_i(\pi_{\sigma,\phi_i}) - J_i(\pi_\sigma) = (J_i(\pi_{\sigma,\phi_i}) - J_i(\pi_{\sigma_E,\phi_i})) + (J_i(\pi_{\sigma_E,\phi_i})) - J_i(\pi_{\sigma_E})) + (J_i(\pi_{\sigma_E}) - J_i(\pi_\sigma))$$
$$\leq J_i(\pi_{\sigma_E,\phi_i}) - J_i(\pi_{\sigma_E}) + O(\epsilon u H)$$

$$(14)$$

Taking the maximum over $i, \phi_i$, we get

$$\mathcal{R}_\Phi(\sigma) - \mathcal{R}_\Phi(\sigma_E) \leq O(u\epsilon H)$$

$\square$

### E.12 Proof of Theorem 5.6

*Proof.* We prove the theorem by constructing such a Markov Game policies that `MALICE` can get $\Omega(\epsilon u H)$ regret gap. We consider a single-agent MDP shown in Figure 4. The rewards are action free. Let $u' = \lfloor u \rfloor$, the reward function $r(s_1) = r(s_3) = ... = r(s_{2u'-3}) = 1$, with all other states yielding a reward of $0$. The transition of the MDP is shown in Figure 4. We know that the value is between $[0, u']$ for any policy, and thus Assumption 5.1 is satisfied.

Let $\sigma_E$ be the policy that $\sigma_E(a_1|s_0) = 1 - \beta, \sigma_E(a_2|s_0) = \beta$. Action at all other states doesn't matter because the transition and the reward would be the same. It is easy to verify that $\sigma_E$ satisfies Assumption 5.2.

Let trained policy $\sigma$ be the policy that $\sigma(a_1|s_0) = 1 - \beta - H\epsilon, \sigma(a_2|s_0) = \beta + H\epsilon$. $\sigma$ and $\sigma_E$ only differ at $s_0$.

Now we verify that $\ell_{\text{TV,MALICE}}(\sigma, D_E, \sigma) \leq \epsilon$.

Since $\sigma$ and $\sigma_E$ only differ at state $s_0$, and $d^{\pi_{\sigma,\phi_i}}(s_0) = 1$ for any $i, \phi_i$, we have that

$$\mathbb{E}_{s \sim d^{\pi_{\sigma_E}}} \left[ \frac{d^{\pi_{\sigma,\phi_i}}}{d^{\pi_{\sigma_E}}} \text{TV}(\sigma_E(s), \sigma(s)) \right] = \mathbb{E}_{s \sim d^{\pi_{\sigma,\phi_i}}} \left[ \text{TV}(\sigma_E(s), \sigma(s)) \right] \leq \frac{1}{H} \cdot H\epsilon = \epsilon$$

Therefore,

$$\ell_{\text{TV,MALICE}}(\sigma, D_E, \sigma) = \max_{i \in [m]} \max_{\phi_i} \mathbb{E}_{s \sim d^{\pi_{\sigma_E}}} \left[ \frac{d^{\pi_{\sigma,\phi_i}}}{d^{\pi_{\sigma_E}}} \text{TV}(\sigma_E(s), \sigma(s)) \right] \leq \epsilon$$

It is not hard to verify, the worst deviation for $\pi_E$ is to deviate action on $s_0$ from playing $a_2$ to $a_1$, and thus

$$\mathcal{R}_\Phi(\pi_E, r) = \beta(u' - 1)$$

the worst deviation for $\pi_E$ is also to deviate action on $s_0$ from playing $a_2$ to $a_1$.

$$\mathcal{R}_\Phi(\pi, r) = (\beta + \epsilon H)(u' - 1)$$

Therefore, the regret gap $\mathcal{R}_\Phi(\pi) - \mathcal{R}_\Phi(\pi_E) = \epsilon(u' - 1)H = \Omega(\epsilon u H)$. $\square$

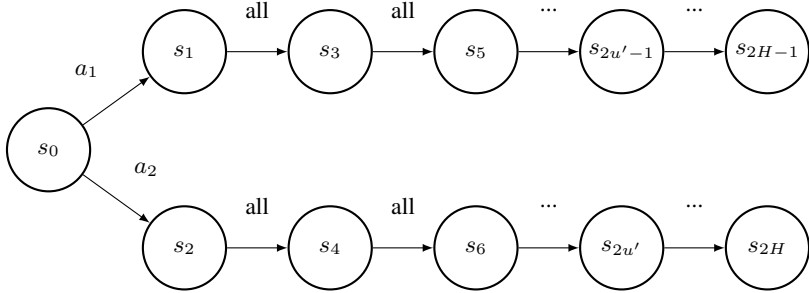

Figure 4: Example of $\Omega(\epsilon u H)$ regret gap for MALICE and BLADES

### E.13 Proof of Theorem 5.7

*Proof.* From the definition of $\ell_{\text{BLADES}}$, we know

$$
\begin{aligned}
\ell_{\text{BLADES}}(\sigma, \sigma) &= \max_{i \in [m]} \max_{\phi_i} \mathbb{E}_{s \sim d^{\pi_\sigma, \phi_i}} \left[ \ell(\sigma_E(s), \sigma_{(}s)) \right] \\
&\geq \max_{i \in [m]} \max_{\phi_i} \mathbb{E}_{s \sim d^{\pi_\sigma, \phi_i}} \left[ \ell(\pi_{\sigma_E, \phi_i}(s), \pi_{\sigma, \phi_i}(s)) \right]
\end{aligned}
\tag{15}
$$

From Lemma C.1, we know that for all $i, \phi_i$, we have

$$
J_i(\pi_{\sigma, \phi_i}) - J_i(\pi_{\sigma_E, \phi_i}) \leq O(\epsilon u H)
$$

And

$$
J_i(\pi_{\sigma_E}) - J_i(\pi_\sigma) \leq O(\epsilon u H)
$$

Therefore, we get

$$
\begin{aligned}
J_i(\pi_{\sigma, \phi_i}) - J_i(\pi_\sigma) &= (J_i(\pi_{\sigma, \phi_i}) - J_i(\pi_{\sigma_E, \phi_i})) + (J_i(\pi_{\sigma_E, \phi_i})) - J_i(\pi_{\sigma_E})) + (J_i(\pi_{\sigma_E}) - J_i(\pi_\sigma)) \\
&\leq J_i(\pi_{\sigma_E, \phi_i}) - J_i(\pi_{\sigma_E}) + O(\epsilon u H)
\end{aligned}
\tag{16}
$$

Taking the maximum over $i, \phi_i$, we get

$$
\mathcal{R}_\Phi(\sigma) - \mathcal{R}_\Phi(\sigma_E) \leq O(\epsilon u H)
$$

$\square$

### E.14 Proof of Theorem 5.8

*Proof.* Let MDP, expert policy $\sigma_E$ and the trained policy $\sigma$ be the same example in the proof of Theorem 5.6.

Since $\sigma$ and $\sigma_E$ only differ at state $s_0$, and $d^{\pi_\sigma, \phi_i}(s_0) = 1$ for any $i, \phi_i$, we have

$$
\mathbb{E}_{s \sim d^{\pi_\sigma, \phi_i}} \left[ \text{TV}(\sigma_E(s), \sigma(s)) \right] \leq \frac{1}{H} \cdot H\epsilon = \epsilon
$$

Therefore, the trained policy $\pi$ satisfies

$$
\ell_{\text{TV,BLADES}}(\sigma, \sigma) = \max_{i \in [m]} \max_{\phi_i} \mathbb{E}_{s \sim d^{\pi_\sigma, \phi_i}} \left[ \text{TV}(\sigma_E(s), \sigma(s)) \right] \leq \epsilon
$$

The regret gap $\mathcal{R}_\Phi(\sigma) - \mathcal{R}_\Phi(\sigma_E) = \epsilon(u' - 1)H = \Omega(\epsilon u H)$. $\square$

## F  Comparison with Goktas et al. [8]

Recent work Goktas et al. [8] worked on similar problem as ours. We will highlight some of the difference between two works.

First, the learning goals are different. They focus on a problem of inverse game theory, where the goal is to recover a reward function to rationalize the expert's behavior, i.e. the expert policy plays

| $r$ | $a_1$ | $a_2$ | $r'$ | $a_1$ | $a_2$ |
|-----|-------|-------|------|-------|-------|
| $a_1$ | (1,1) | (0,0) | $a_1$ | (1,1) | (0,0) |
| $a_2$ | (0,0) | (2,2) | $a_2$ | (0,0) | (1,1) |

Figure 5: Multiple reward functions rationalize $\sigma_E$

an equilibrium under such a reward function. However, in our setting, instead of recovering a singe reward function, our goal is to learn a robust policy that get similar regret performance under a class of reward functions. We will show later that if the ultimate goal is to learn this robust policy, simply recovering a single reward function is not enough.

Second, the solution concepts are different. they work on Nash equilibrium, while in our setting, we focus on correlated equilibrium. We note that our algorithms also work for learning independent policies, by restricting the policy class to be a class of independent policies.

Third, in finite demonstration setting, their objective is to find a reward function which the learned policy plays a local NE, under the constraints that $\ell_2$ difference of the observations for behaviors of two learned policy is small. We note that in general simply matching this difference is not enough to guarantee that the learned policy play an equilibrium. From Theorem 4.3, we know that even if the occupancy measures of two policies exactly match, the regrets can still be significantly different under the same reward function.

In conclusion, they work on a inverse game theory style problem where the goal is to recover a single reward function to rationalize the agents behavior. We work on imitation learning problem, where the goal is not recovering a single reward function but learning a policy that matches the regret performance of the expert under a class of reward functions.

We will give examples in normal form games (NFG) to show that recovering a single reward function is not enough to learn a policy that minimizing the regret gap for a large class of reward functions. NFG can be viewed as an MG in which $H = 1$ and $|\mathcal{S}| = 1$.

**Lemma F.1.** *For an expert policy $\sigma_E$, there may exist multiple reward functions that rationalize it.*

*Proof.* We show this by an example of normal form games in Figure 5. Consider the policy to be $\sigma_E(a_1 a_1) = 1$, then the expert plays CE/NE under both reward functions $r$ and $r'$, which means both reward functions rationalize $\sigma_E$. □

**Lemma F.2.** *For a fixed reward function, There may exist multiple CE/NEs.*

*Proof.* For reward function $r$ in Figure 5, we can construct such two policies $\sigma_1, \sigma_2$. For $\sigma_1$, let $\sigma_1(a_1, a_1) = 1$. Let $\sigma_2(a_1, a_1) = \frac{4}{9}, \sigma_2(a_1, a_2) = \sigma_2(a_2, a_1) = \frac{2}{9}, \sigma_2(a_2, a_2) = \frac{1}{9}$. Tt is not hard to verify that both $\sigma_1$ and $\sigma_2$ play CE/NE under the reward function $r$. □

Therefore, since there is no one-to-one mapping between the equilibria and the pay-off structures, simply recovering a single reward function might not help recover a policy that gets small regret gap.

For example, the true reward function is $r$ in Figure 5, and expert policy $\sigma_E$ satisfies $\sigma_E(a_1, a_1) = 1$. The algorithm may recover $r'$ in Figure 5, and a trained policy $\sigma$ that plays NE/CE under recovered reward function $r'$ would be $\sigma(a_1, a_1) = \sigma(a_1, a_2) = \sigma(a_2, a_1) = \sigma(a_2, a_2) = \frac{1}{4}$. However, this trained policy $\sigma$ does not play NE/CE under the true reward function $r$.

