# OpenReview forum: "Multi-Agent Imitation Learning: Value is Easy, Regret is Hard"
_NeurIPS.cc/2024/Conference — NeurIPS 2024 poster_

### Official Review · Reviewer_SJkf · 2024-07-12

**Soundness:** 3
**Presentation:** 2
**Contribution:** 3
**Rating:** 6
**Confidence:** 2

**Summary:**

The paper studies the problem of an mediator coordinating the actions of a collection of agents by learning from the demonstrations of an expert. Unlike previous work the paper considers strategic deviations of the agents given the recommended actions instead of just matching the experts recommendations. It is shown the minimizing the standard performance measure (value gap) can lead to high regret in this setting. Finally, two algorithms are shown to minimize the regret under some assumptions.

**Strengths:**

1-The paper makes a nice extension of prior work. It is reasonable to expect the agents to deviate to maximize their utility instead of just following the recommended actions.

2-I am not familiar with this area and the related work, but the paper has a nice logical flow. It starts by showing the difference between the performance measure of value and regret. Then it gives two algorithms to minimize the regret.

**Weaknesses:**

1-The paper's presentation can be improved. So many theorems are followed by [proof] in blue which looks very sloppy.

2-It seems that the motivation of the paper has some similarities with incentivized exploration where the principle recommends an action but the agent is not forced to follow it, but I did not see it mentioned in the related work. Perhaps, there is even an extension of this work that connects to incentivized exploration.

**Questions:**

1-In lines (123-124) it says "Critically, no agent observes the recommendations the mediator provides to another agent", but in the definition of regret between lines 50 and 51 we have the agent maximizing utility based on $\pi_{\sigma,\phi_i}$ which is the "joint agent policy induced by all agents other than i following σ’s recommendations". I don't understand how agent $i$ can find the strategy to maximize utility if he does not know the recommended actions and therefore  $\pi_{\sigma,\phi_i}$ ?


2-Please see the points above under Weaknesses.

**Limitations:**

I think the limitations were properly addressed.

---

> ### Author Rebuttal · Authors · 2024-08-06
>
> Thank you for taking the time to review our submission! Please find our replies to your questions/comments below.
>
> Response to W1: We designed the [proof] link after each theory statement to add a hyperlink to the proof in the appendix. This allows readers to easily navigate from the theorem statements to the corresponding proofs and back from the proofs to the theorems. We will follow the reviewer’s suggestion and make it clear in the final version by adding “detailed proof can be found in Sec X in Appendix” along with the hyperlink.
>
> Response to W2: We agree that incentivized exploration is similar to our problem in that agents can choose to deviate from the recommendations. However, the goal of MAIL in this paper is to learn a mediator policy from an expert mediator to coordinate the agents. In contrast, the goal of incentivized exploration is to provide recommendations to agents to incentivize exploration rather than exploitation, where there is no access to an expert mediator.
>
> Answer to Q1: The goal of minimizing the regret gap in this paper is to learn a mediator who coordinates the agents, ensuring that the agents have no incentive to deviate from the recommendations. This means that we focus on learning the mediator's policy, not the agents' best response policy. By minimizing the regret gap, we ensure that the maximum additional gain agents can achieve with any possible deviation is similar to the additional gain they would get by deviating from the expert mediator. Regarding the agents, they can maximize their utility by treating other agents as part of the environment, and set the state as $(s,a)$, where $a$ is the recommendation from the mediator, and use a reinforcement learning algorithm to learn a mapping from $S\times A\rightarrow A$.

---

### Official Review · Reviewer_aTBa · 2024-07-13

**Soundness:** 2
**Presentation:** 2
**Contribution:** 2
**Rating:** 5
**Confidence:** 2

**Summary:**

In this work, the authors 1.initiate the study of an alternative objective for multi-agent imitation learning termed the regret gap; 2. investigate the relationship between regret gap and the value gap; 3. provided 2 provably efficient algorithms to minimize the regret gap under coverage assumption on the expert or access to a queryable expert respectively.

**Strengths:**

This work proposes to study a new alternative objective for multi-agent imitation learning, which is one step closer to downstream applications like routing a group of drivers.

The paper contains both upper and lower bounds for regret gap optimization, along with detailed proofs.

The presentation is generally clear, and the appendix is well-organized.

**Weaknesses:**

Some assumptions are not explicitly made clear, such as the realizable expert, which can be relatively strong in practice. Also, in line 123, the requirement that no agent observes the recommendations the mediator provides to another agent sounds restrictive. Additionally, in line 146, it is required that identity mapping is in the class. It would be beneficial to have an extensive summary of assumptions in the appendix.

Proposed algorithms seems like straight forward applications of existing algorithms, it would be nice to list the challenges when applying existing results.

No experimentation or code is included with the work, making it difficult to examine whether the proposed algorithms are efficient in practice.

**Questions:**

Q1: The notion of regret gap seems confusing to me. Is it related to the regret in online learning? Could you please explain the reasoning behind this naming?

Q2: I assume realizability (i.e., the expert policy being contained in the learner's policy class) is required for your theorems to hold. Is that true?

Q3: It would be helpful if the authors could state the challenges encountered when obtaining new theoretical results based on previous works.

Q4: It would be helpful if there is an experimental plan to verify the algorithms, such as in simulated driver routing games.

Minor suggestions:
Is there a typo in line 147 that begins with "identity mapping ..."? Should that sentence end with "Φi"?

**Limitations:**

Same as weakness. Realizability is a limitation. As stated in limitations, this work focus on tabular MGs, while the computation and sample/query complexity is linear over the cardinality of the deviation class.

---

> ### Author Rebuttal · Authors · 2024-08-06
>
> Thank you for taking the time to review our submission! Please find our replies to your questions/comments below.
>
> Regarding the assumptions:
>
> 1. The theorems in this paper do not require realizability to hold. However, if we want to further analyze the computational complexity of the algorithm (which would be an additional theorem not stated in the paper), realizability can simplify the analysis, as discussed in single-agent imitation learning (see Appendix B.3 of [21]).
>
> 2. In game theory literature, it is common for agents to observe only the recommendations provided to themselves in the study of (coarse) correlated equilibrium. This scenario is also typical in practice. For example, a route planning application provides recommendations to many drivers, but each driver only receives their own recommendation. It is reasonable that they cannot see whether the app advises other drivers to turn left or go straight.
>
> 3. Identity mapping lies in the class is not an assumption, in the sense that it simply means that the agent can choose to fully follow the recommendation of the mediator.
>
> Regarding the novelty of the algorithms: Our proposed algorithms are indeed built on the previous approaches taken by algorithms like DAgger and ALICE (as the names suggest). However, we have to develop a non-trivial extension of these algorithms for minimizing the regret gap. While the natural extensions of DAgger and ALICE are able to minimize the value gap in MAIL, they fail to minimize the regret gap. To minimize the regret gap, it is necessary to visit the counterfactual states visited by the deviated policy, and this aspect of exploration is our novel contribution.
>
> Regarding the experiments: We choose to focus on theory because we believe there are substantial basic theoretical questions that remain unanswered. Our primary contribution is to advance the theoretical understanding of the objective formulation for MAIL and its implications for algorithm design, which, to the best of our knowledge, have not been addressed by prior works.
>
> Answer to Q1: We adopt this naming from [27], where 'regret' is defined in normal form games (NFG), the standard matrix game in the game theory literature, which can be viewed as an one step ($H=1$) markov game with only a single state. The term 'regret' does not directly relate to the concept of regret in online learning. This naming aims to capture the agent's 'regret' for not deviating from the mediator's recommendations.
>
> Answer to Q2: As mentioned earlier, the theorems in this paper do not require realizability to hold. However, if we want to further analyze the computational complexity of the algorithm (which would be an additional theorem not stated in the paper), realizability can simplify the analysis, as discussed in single-agent imitation learning (see Appendix B.3 of [21]).
>
> Answer to Q3: Thank you for your interest in the challenges we encountered during this project. The biggest challenge is that we found out without certain assumptions, all natural extensions of the single-agent IL algorithms (e.g. DAgger, IRL algorithms, BC) can only minimize the value gap, but fail to provide any guarantees for minimizing the regret gap. That means we cannot get any theoretical results for any of those algorithms and have to adopt new assumptions and develop new algorithms to minimize the regret gap.
>
> Answer to Q4: As mentioned earlier, we choose to focus on theory because we believe there are substantial basic theoretical questions that remain unanswered. Our primary contribution is to advance the theoretical understanding of the objective formulation for MAIL and its implications for algorithm design, which, to the best of our knowledge, have not been addressed by prior works.
> As for the algorithms, we believe the theoretical results provide insights into designing a practical MAIL algorithm. The key takeaway is that simply extending a single-agent IL algorithm straightforwardly is insufficient to minimize the regret gap (as all previous empirical works have done, to the best of our knowledge). In the multi-agent setting, additional efforts are needed to minimize the regret gap. This includes interactively querying the expert on counterfactual states or applying importance weighting to the expert demonstrations based on the state distribution induced by the deviated policies.
>
> For Minor Suggestions: Thank you for pointing that out. We will fix this typo.

---

### Official Review · Reviewer_NThg · 2024-07-13

**Soundness:** 3
**Presentation:** 4
**Contribution:** 2
**Rating:** 6
**Confidence:** 3

**Summary:**

This paper addresses the challenge of a mediator coordinating a group of strategic agents by recommending actions. Unlike previous work that focuses on non-strategic users who follow recommendations blindly, this study explores strategic users who may deviate based on their personal utility functions. The authors find that the traditional approach of minimizing the value gap is insufficient for handling deviations by strategic agents, as it fails to account for recommendations outside the induced state distribution. To address this, the paper introduces the concept of the regret gap in multi-agent imitation learning (MAIL) within Markov Games, which explicitly considers potential deviations by agents. The authors demonstrate that under the assumption of complete reward and deviation function classes, achieving regret equivalence ensures value equivalence. However, value equivalence does not guarantee a small regret gap. To minimize the regret gap, the paper presents two efficient algorithms: MALICE, which operates under coverage assumptions, and BLADES, which requires access to a queryable expert. These contributions offer a novel perspective and tools for improving robustness in multi-agent coordination scenarios involving strategic agents.

**Strengths:**

1. **Practical Relevance**: The paper addresses a significant real-world problem in multi-agent imitation learning, applicable in scenarios such as routing a group of drivers through a road network.
2. **Comprehensive Literature Review**: The literature review is thorough, contrasting the prevalent value gap-based approaches in empirical MAIL work with the novel focus on regret gap-based methods.
3. **Strong Motivation**: The paper is well-motivated, extending the existing framework to include strategic users and demonstrating the limitations of previous methods, thereby justifying the need for their proposed approach.
4. **Clarity and Readability**: The paper is well-written and easy to follow, making complex concepts accessible to the reader.
5. **Discussion of Assumptions and Limitations**: The authors provide a clear discussion of the assumptions and limitations of their work, adding depth to their analysis.

**Weaknesses:**

1. **Algorithmic Novelty**: The proposed algorithms, MALICE and BLADES, may lack novelty. MALICE modifies behavior cloning with an importance sampling term, relying on the expert visiting every state with positive probability. BLADES involves querying the expert for action recommendations based on possible deviations. These approaches are not entirely new.
2. **Lack of Empirical Evaluation**: The algorithms have not been empirically evaluated, even in tabular Markov games. Their actual performance remains unknown, and the requirement to enumerate all user deviations may be impractical for real-world applications.

**Questions:**

1. What is the core difference between Algorithm 1 and Algorithm 2? They both enumerate all possible user deviations. Then Algorithm 1 requires a full demonstration coverage assumption to ensure that the importance weights are finite, and Algorithm 2 requests the expert to provide recommendations under all possible user deviations. These two requirements seem to be similar. Could the authors provide more discussions about their core differences?

2. How strict is the Assumption 5.1 (u-recoverability)? This assumption states that all single-agent deviations only incur bounded advantage fluctuations, which seems to be too strict.

3. Is Figure 1 a good example? The Markov game in Figure 1 assumes that the expert only visits the states in the lower path, which contradicts Assumption 5.2 ($\beta$-coverage), I understand that Figure 1 is used to show why "regret is hard", but it is still confusing.

**Limitations:**

The authors have adequately addressed the limitations of the proposed algorithms.

---

> ### Author Rebuttal · Authors · 2024-08-06
>
> Thank you for taking the time to review our submission! Please find our replies to your questions/comments below.
>
> Response to W1: Our proposed algorithms are indeed built on the previous approaches taken by algorithms like DAgger and ALICE (as the names suggest). However, we have to develop a non-trivial extension of these algorithms for minimizing the regret gap. While the natural extensions of DAgger and ALICE are able to minimize the value gap in MAIL, they fail to minimize the regret gap. To minimize the regret gap, it is necessary to visit the counterfactual states visited by the deviated policy, as shown in lines 5-7 of both Algorithm 1 and Algorithm 2, and this aspect of exploration is our novel contribution.
>
> Response to W2: We choose to focus on theory because we believe there are substantial basic theoretical questions that remain unanswered. Our primary contribution is to advance the theoretical understanding of the objective formulation for MAIL and its implications for algorithm design, which, to the best of our knowledge, have not been addressed by prior works.
> As for the algorithms, we believe the theoretical results provide insights into designing a practical MAIL algorithm. The key takeaway is that simply extending a single-agent IL algorithm straightforwardly is insufficient to minimize the regret gap (as all previous empirical works have done, to the best of our knowledge). In the multi-agent setting, additional efforts are needed to minimize the regret gap. This includes interactively querying the expert on counterfactual states or applying importance weighting to the expert demonstrations based on the state distribution induced by the deviated policies.
>
> Answer to Q1: Although the theoretical proofs of BLADES and MALICE are similar, they represent two distinct paradigms of IL algorithms. To run BLADES, you will need to have access to a queryable expert (similar to DAgger), whereas to run MALICE, with a coverage assumption, you only need to use importance weighting on expert demonstrations without requiring additional interaction with the expert.
>
> Answer to Q2: As stated in line 224, a small value of $u$ indicates that we are not in a scenario where a one-time joint mistake occurs, and a single agent deviates, and even the (deviated) expert couldn’t recover for the rest of the episode. The recoverability assumption is common in single-agent imitation learning (SAIL), as seen in DAgger [18] and ALICE [20], and ours can be viewed as a natural extension to the multi-agent setting. We believe that in many cases $u$ is small. For instance, in the route planning example, at some point many cars may miss their turns/intersections/exits, but this can be recovered within a constant time, even when a single agent chooses not to follow its recommendation, rather than $u$ increasing as $H$ increases.
>
> Answer to Q3: Figure 1 illustrates the dynamics, while Assumption 5.2 is about the expert policy. For further details, you could check the proof of Theorem 5.2 in Appendix E.7. The example in the proof satisfies $\beta$-coverage, and the dynamics is the same as it is in Figure 1.

---

> > ### Comment · Reviewer_NThg · 2024-08-14
> >
> > Thank you for your clarification. I increase the score to 6.

---

### Official Review · Reviewer_3HJn · 2024-07-15

**Soundness:** 3
**Presentation:** 3
**Contribution:** 2
**Rating:** 5
**Confidence:** 3

**Summary:**

The paper studies the problem of multi-agent imitation learning (MAIL), with a focus on the differences and relationships between the value gap objective and the regret gap objective.

The set-up consists of a multi-agent Markov game, where a central mediator attempts to coordinate the behaviors of the agents. Two potential objectives are compared: (1) Value gap objective: the coordinator tries to minimize the maximum deficiency among the agents in the overall value under the (joint) learning policy compared to the expert policy; (2) Regret gap objective: the coordinator minimizes instead the excess regret with respect to the expert policy, with regret defined as the maximum utility gain among the agents by deviating from the recommendation.

The authors show that, under certain completeness assumptions of both the reward and the deviation classes, regret equivalence implies value equivalence; In contrast, value equivalence in general does not imply regret equivalence. These are in sharp contrast to the single-agent imitation learning setting, where the two notions become equivalent.

While regret equivalence is hard in general, the authors provide two algorithms, MALICE and BLADES, under different assumptions on recoverability, policy coverage, and/or query-ability of expert. Both algorithms provide a linear (in horizon) bound on the regret gap, which matches the single agent case.

**Strengths:**

[Originality and significance] The paper brings novel insights on MAIL in the sense that it characterizes how MAIL differs from SAIL in the regret perspective. Both the value-regret relationship results and the algorithms demonstrate the hardness in the multi-agent case due to lack of access to expert recommendations in counter-factual settings. In addition to the hardness results, the authors also provide algorithms that can be potentially useful to tackle the regret gap problem; this is valuable despite the still limited conditions.

[Quality and clarity] The paper is generally well written and easy for me to follow. The results and their proofs seem correct.

**Weaknesses:**

The main limitations of this paper lie in the significance of the results, both from the theoretical and the practical front. This is by no means to diminish the value of the results. What concerns me is two folds: (1) The main claims on the hardness of value gap and regret gap see mostly standard. The hardness due to inability to observe certain information in the multi-agent setting seems generally intuitive and unsurprising, and I think prior works (e.g., [1]) have to some extent demonstrated similar insights in related (albeit different) settings. The contribution seems more marginal in this respect. (I do perceive the contribution on the algorithmic side, though, as more substantial.) (2) On the practical side, I am not sure how the proposed methodology shall be applied to solving real world problems. Despite the theoretical nature of this paper, additional discussion on this can help make the paper more approachable to a larger audience.

[1] Zhao, G., Zhu, B., Jiao, J., & Jordan, M. Online learning in Stackelberg games with an omniscient follower. International Conference on Machine Learning 2023.

**Questions:**

- Line 172: I am not sure why Theorem 4.2 “proves that the completeness of the classes is *necessary*”. I suspect the word “necessary” is not in its mathematical meaning, since the theorem seems to only show that some condition on the classes is necessary.
- Line 179: Why is this the italicized statement “surprising”? I in fact find it quite expected, especially given the discussion in Line 176-177 on the regret equivalence as a stronger notion. This really makes me think that I’m missing on some key intuition regarding why one could hope for the direct implication.
- I would like to see more discussion on how restrictive the assumptions (in particular, the recoverability Assumption 5.1) actually are, e.g., when one could expect them to hold. This would help better bridge the gap between the theoretical results and the practice.

Minor comments:
- Line 146: I find this notation a bit unstandard $\Psi := \{\Psi_i\}_{I=1}^m$ (as a sequence of sets). Maybe the product is more appropriate?
- I think $r$ and $f$ are used interchangeably when denoting the reward function. Nothing wrong, but I suppose they could be made consistent.
- Line 183: Typo “even” => “even if”

**Limitations:**

Addressed in Section 6 and Appendix A. The paper is of a theoretical nature.

---

> ### Author Rebuttal · Authors · 2024-08-06
>
> Thank you for taking the time to review our submission! Please find our replies to your questions/comments below.
>
> Regarding the significance of theoretical results: We thank the reviewer for bringing up paper [1]. [1] shows that in the repeated Stackelberg game, not being able to observe the follower's response will result in higher regret. At a very high level, one can say that the hardness in MAIL in this paper comes from the inability of observing the expert’s recommendations at counterfactual states if any agent deviates. However, we would like to emphasize that the fundamental reasons are very different from [1]: the hardness of minimizing the regret gap in MAIL mainly comes from the counterfactual states, that is, the state distribution shift brought by deviations. This only becomes a problem when (1) there are multiple agents instead of a single agent, and (2) the game is a Markov Game with multiple states and a long horizon. (For normal form games where $|S|=H=1$, [27] can efficiently minimize the regret gap.) In [1] that the reviewer refers to, the setting is a repeated Stackelberg game, which does not have similar issues. Additionally, the setting is different in that we aim to learn a mediator policy given the expert mediator, while that paper aims to achieve regret minimization for players in a cooperative game. Furthermore, in MAIL, we are the first to point out such hardness related to the regret gap, which has been ignored by the MAIL community for a long time. For these reasons, we argue that the theoretical contribution of this paper is substantial and the significance is not impacted by the results in [1]. We are open to exploring the connection if the reviewer can elaborate further.
>
> Regarding practical implications: We choose to focus on theory because we believe there are substantial basic theoretical questions that remain unanswered. Our primary contribution is to advance the theoretical understanding of the objective formulation for MAIL and its implications for algorithm design, which, to the best of our knowledge, have not been addressed by prior works.
> As for the algorithms, we believe the theoretical results provide insights into designing a practical MAIL algorithm. The key takeaway is that simply extending a single-agent IL algorithm straightforwardly is insufficient to minimize the regret gap (as all previous empirical works have done, to the best of our knowledge). In the multi-agent setting, additional efforts are needed to minimize the regret gap. This includes interactively querying the expert on counterfactual states or applying importance weighting to the expert demonstrations based on the state distribution induced by the deviated policies.
>
> Answer to Q1: Thank you for pointing that out. The word "necessary" is used in its natural language sense and not in its mathematical meaning. We will fix that to avoid confusion.
>
> Answer to Q2: We use the word "surprising" to highlight our findings:
> 1. Regret gap and value gap are the same in the single-agent setting (see Appendix D), but value equivalence does not imply regret equivalence in the multi-agent setting.
> 2. Not only does value equivalence not directly imply regret equivalence, but it also cannot provide any guarantees for the regret gap, which can be *arbitrarily large*.
>
> Answer to Q3: As stated in line 224, a small value of $u$ indicates that we are not in a scenario where a one-time joint mistake occurs, and a single agent deviates, and even the (deviated) expert couldn’t recover for the rest of the episode. The recoverability assumption is common in single-agent imitation learning (SAIL), as seen in DAgger [18] and ALICE [20], and ours can be viewed as a natural extension to the multi-agent setting. We believe that in many cases $u$ is small. For instance, in the route planning example, at some point many cars may miss their turns/intersections/exits, but this can be recovered within a constant time, even when a single agent chooses not to follow its recommendation, rather than $u$ increasing as $H$ increases.
>
> Regarding Minor Comments:
>
> [1] In the standard game theory notion, a correlated equilibrium (CE) considers the deviation of a single agent and thus does not require defining deviations as a product of individual deviations. Therefore, we use the current notation to avoid confusion.
>
> [2] We inherit this notation from the single-agent IL literature, where they usually let $r$ denote the true reward function, and $f$ denotes any possible reward function in the reward function class.
>
> [3]Thank you for pointing this out. We will fix this typo.

---

### Decision · Program_Chairs · 2024-09-25

**Decision:**

Accept (poster)

**Comment:**

This paper considers the problem of imitation learning in multi-agent settings, and provides basic theoretical foundations, including:
1) They compare two potential objectives, the value gap and regret gap, and show that while regret equivalence implies value equivalence, value equivalence does not imply regret equivalence in general.
2) While regret equivalence can be harder to achieve in general, they new provide algorithms that achieve this under additional assumptions (either coverage or access to a queryable expert).

Reviewers found the problem this paper studies to be well-motivated and important, and generally felt that the paper makes a reasonable first attempt at providing basic foundations for multi-agent IL. They also found the paper to be clear and well-written, with a nice and relatively complete story. The main limitation highlighted by the reviewers was limited technical depth/surprise factor in the main results. I believe this is OK, given that the theoretical formulation is itself novel and important, but for the final revision I recommend better highlighting the technical challenges to the extent that this is possible.